# GRASP: Awakening Latent Spatial Reasoning in LVLMs via Training-free Geometric Rectification

**Jiadong Yan**[1]  **Ke Zhang**[1]  **Chenyang Zhao**[1]  **Shoushan Li**[1]  **Xizhao Luo**[1]

## Abstract

Large Vision-Language Models (LVLMs) exhibit remarkable general capabilities but struggle significantly with spatial reasoning tasks. In this paper, we uncover a critical representation-output misalignment via linear probing: LVLMs correctly encode spatial features internally, but generate incorrect results in the final text. To address this, we pioneer the Inference-time Geometric Manifold Adaptation paradigm and propose **GRASP** (**G**eometric **R**ectification for **A**ctive **S**patial **P**erception), a training-free framework to awaken these latent capabilities. GRASP employs Manifold Differential Search to identify optimal geometric counterfactuals, which then drive a dual-level rectification mechanism: Implicit Trajectory Correction to rectify attenuated intrinsic geometric features in intermediate decoder layers, and Explicit Distribution Alignment to break the dominance of language priors at the output layer. Extensive experiments spanning diverse architectures (LLaVA, Qwen 2.5/3-VL) and positional encoding paradigms (1D APE, 2D/3D RoPE) across image and video benchmarks (WhatsUp, VSR, VSI-Bench) demonstrate that GRASP significantly mitigates spatial hallucinations without parameter updates, achieving accuracy gains of up to 26.1% on image benchmarks and 9.7% on video reasoning tasks, consistently outperforming baseline methods.

## 1. Introduction

The rapid evolution of Large Vision-Language Models (LVLMs) (Dai et al., 2023; Bai et al., 2023; Zhu et al., 2023; Liu et al., 2024b) has revolutionized general visual recognition (Jiang et al., 2024; Wu et al., 2025c), yet these models exhibit a persistent vulnerability in spatial reasoning (Xu et al., 2025; Zheng et al., 2025). They frequently generate directional hallucinations that contradict visual evidence. This deficiency severely limits their precise perception of the physical world and hinders deployment in safety-critical applications such as autonomous driving (Tian et al., 2024) and embodied AI (Zitkovich et al., 2023).

To address this challenge, prior works have relied on data-intensive Supervised Fine-Tuning (SFT) (Liu et al., 2024a) or inference-time reasoning enhancement via Chain-of-Thought (CoT) (Zhang et al., 2023) and Reinforcement Learning (RL) (Yu et al., 2024). However, SFT struggles with open-world generalization and recent findings (Yang et al., 2025b; Liao et al., 2025) indicate that without targeted RL, complex CoT strategies often underperform vanilla generation in spatial reasoning tasks. This suggests that the root cause of spatial hallucination is not a deficiency in high-level logical reasoning, but rather a misalignment between low-level visual evidence and high-level semantic generation (Tong et al., 2024). Although recent mechanistic studies (Li et al., 2025a; Zhang et al., 2025b) have identified the vision encoder's positional encodings as a decisive factor for spatial understanding, they are predominantly limited to static analysis or adjustments, failing to actively leverage the latent geometric capabilities of positional encodings. However, to effectively activate these capabilities, we must first diagnose the precise origin of the spatial hallucination.

This raises a fundamental question: Does the model's failure stem from an inability to *perceive* geometry, or a failure to *express* it? To answer this, we employ Linear Probing (Alain & Bengio, 2016) and Logit Lens (Belrose et al., 2023) as diagnostic tools to dissect the model's internal states. Our investigation on both In-Distribution (ID) and Out-of-Distribution (OOD) datasets reveals a counter-intuitive phenomenon of representation-output misalignment: **(1) Precise Internal Encoding.** Intermediate decoder layers exhibit high spatial discriminability that significantly exceeds the model's final performance. As shown in Figure 1 (Top), the ID probing accuracy for LLaVA-1.5 (Liu et al., 2024a) ascends to near 100% in intermediate layers and maintains this high plateau until the final layer. Even on

[1]School of Computer Science and Technology, Soochow University, Suzhou, China. Correspondence to: Ke Zhang <kzhang19@suda.edu.cn>.

*Proceedings of the 43rd International Conference on Machine Learning*, Seoul, South Korea. PMLR 306, 2026. Copyright 2026 by the author(s).

unseen OOD samples, the probing accuracy at peak layers reaches 86%, significantly surpassing its 60.3% final output accuracy. Similar phenomena are consistently observed in the Qwen2.5-VL (Bai et al., 2025) (Figure 1 Bottom). This confirms the model internally *perceives* the correct spatial layout. **(2) Suppression by Language Priors.** These distinct geometric signals fail to persist. While intermediate layers are dominated by the visually correct token as verified by logit lens in Figure 2, OOD probing curves in Figure 1 suffer a precipitous decline after peaking. This suggests that valid spatial features are overridden by language priors as the information flows to deep layers. Essentially, the model correctly encodes the visual truth but fails to *express* it, as the final generation is biased by linguistic habits rather than the visual evidence.

Based on these findings, we propose **GRASP** (**G**eometric **R**ectification for **A**ctive **S**patial **P**erception), a training-free framework that awakens latent spatial capabilities via inference-time geometric manifold adaptation. Premised on the insight that positional encodings fundamentally operate on a Lie Group manifold (Bronstein et al., 2021; Hutchinson et al., 2021; Su et al., 2024), we posit that spatial perception can be actively calibrated by traversing the tangent space of this latent geometry, shifting from passive encoding to Active Spatial Perception. GRASP thus reformulates spatial reasoning as an active trajectory generation. It first executes **Manifold Differential Search** to locate a geometric counterfactual reference inducing maximum semantic conflict. Anchored by this reference, a dual-level rectification awakens latent spatial reasoning: **(1) Implicit Trajectory Correction** injects geometric signals derived from the counterfactual to rectify attenuated geometric features in internal layers; **(2) Explicit Distribution Alignment** utilizes the reference to measure semantic discrepancies, dynamically calibrating output logits to neutralize language priors.

In summary, our main contributions are as follows:

- We identify a **representation-output misalignment** as the root cause of spatial hallucinations through a mechanistic analysis: Linear Probing reveals the contrast between spatial encoding and feature attenuation, while Logit Lens visualizes the semantic collapse where visual signals are suppressed by language priors.

- We pioneer a novel **Inference-time Geometric Manifold Adaptation** paradigm by proposing GRASP. The training-free framework utilizes manifold differential search to awaken latent geometric capabilities, synergizing implicit trajectory correction and explicit distribution alignment to counteract language dominance.

- We demonstrate GRASP's superior robustness across diverse LVLMs and multiple spatial reasoning benchmarks. To the best of our knowledge, this is the **first**

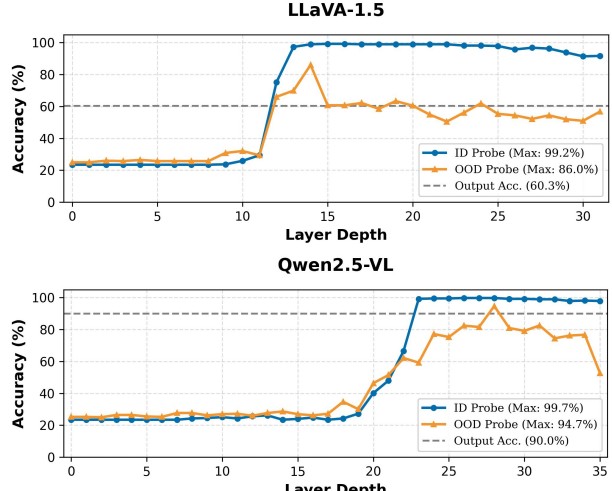

*Figure 1.* **Representation-Output Misalignment.** Finding 1: Internal spatial discriminability ascends to near 100%, presenting a stark contrast to the significantly lower final output. Finding 2: The subsequent steep decline in deep layers quantifies the suppression of these visual signals by language priors.

**unified inference-time intervention framework** applicable to both image and video tasks across distinct positional encoding architectures (1D APE, 2D/3D RoPE).

## 2. Preliminaries: A Geometric Manifold Perspective

In this section, we reformulate LVLM spatial representations as operations on a geometric manifold. We characterize self-attention sensitivity to geometric transformations and analyze how positional encodings enable continuous operators within the feature space.

### 2.1. Self-Attention as a Geometric Function

Given a visual feature sequence $\mathbf{X} \in \mathbb{R}^{N \times D}$, the Transformer aggregates spatial information via self-attention (Vaswani et al., 2017). To inject geometric coordinates $\mathbf{c}_i$ into queries and keys, Absolute Positional Encoding (APE) additively uses $\mathbf{q}_i = \mathbf{W}_q(\mathbf{x}_i + \mathbf{p}(\mathbf{c}_i))$, whereas Rotary Positional Embedding (RoPE) employs a multiplicative formulation $\mathbf{q}_i = \phi_q(\mathbf{x}_i, \mathbf{c}_i)$ and $\mathbf{k}_j = \phi_k(\mathbf{x}_j, \mathbf{c}_j)$. The attention scores $\mathbf{S}_{i,j} \propto \mathbf{q}_i^\top \mathbf{k}_j$ rely on the coordinate system $\mathbf{c}$ due to the permutation invariance of the dot product. We conceptualize the positional encoding as governed by a geometric parameter $\theta$ via a transformation operator $\mathcal{T}_\theta$. Consequently, any perturbation on the coordinate manifold $\mathcal{T}_\theta(\mathbf{c})$ propagates deterministically through the encoding function to reshape the spatial attention distribution $\mathbf{S}(\theta)$, serving as the mathematical basis for our intervention.

## 2.2. Parametrizing the Positional Manifold

Existing LVLMs primarily adopt two positional encoding paradigms. We reveal that both fundamentally underlie a latent geometric manifold, enabling unified differentiable transformations.

**Learnable Absolute Positional Encoding (APE).** Architectures like LLaVA (Liu et al., 2024a) use a learnable matrix $\mathbf{P} \in \mathbb{R}^{N \times D}$. While discrete, $\mathbf{P}$ is topologically homeomorphic to the 2D image grid. We model it as a discretized geometric manifold. Letting $\mathbf{p}(\mathbf{c}_i)$ denote the position vector corresponding to coordinate $\mathbf{c}_i$, the encoding formulation is: $\mathbf{q}_i = \mathbf{x}_i + \mathbf{p}(\mathbf{c}_i)$. From this perspective, we define a resampling operator $\mathcal{W}$. For an affine parameter $\theta$, the transformed encoding is obtained via interpolation: $\mathbf{P}' = \mathcal{W}(\mathbf{P}, \theta)$. This converts discrete parameters into optimizable geometric variables.

**Rotary Positional Embedding (RoPE).** RoPE (Su et al., 2024) employs a multiplicative mechanism, encoding positions via rotations in the complex domain or high-dimensional subspaces. The encoding function is defined as: $\mathbf{q}_i = \mathcal{R}(\mathbf{c}_i)\mathbf{x}_i$, where $\mathcal{R}(\mathbf{c}_i)$ is an orthogonal rotation matrix uniquely determined by coordinate $\mathbf{c}_i$. From a group-theoretic perspective, the core advantage of RoPE lies in its construction of a continuous manifold endowed with a Lie Group structure. Specifically, the rotation matrices $\mathcal{R}$ constitute the special orthogonal group $SO(2)$. Attention scores thus depend solely on relative coordinate differences: $\mathbf{q}_i^T \mathbf{k}_j = \mathbf{x}_i^T \mathcal{R}(\mathbf{c}_j - \mathbf{c}_i)\mathbf{x}_j$. Modern LVLMs (e.g., Qwen-VL (Bai et al., 2025)) extended this to multi-dimensional coordinate $\mathbf{c}$ by decomposing feature subspaces: **(1) 2D RoPE:** For images, the feature dimensions are decomposed into two subspaces, utilizing $\mathcal{R}(h)$ and $\mathcal{R}(w)$ to encode height and width, respectively. **(2) 3D RoPE:** For video tasks, a temporal dimension $t$ is introduced to construct a spatiotemporal coordinate system $\mathbf{c} = (t, h, w)$. Crucially, RoPE exhibits geometric equivariance: a transformation $\mathbf{c}' = \mathcal{T}_\theta(\mathbf{c})$ operates as a group action on the Lie Group manifold, enabling precise spatial control solely by traversing the coordinate space without modifying model weights.

## 3. Why Do LVLMs Hallucinate on Spatial Relationships?

To investigate the mechanistic origins of spatial hallucinations, we delve into the model's internal dynamics to trace the evolution of geometric information during the decoding process. We employ two complementary analytical methods: Linear Probing (Alain & Bengio, 2016), which employs simple external classifiers on internal layers to quantify whether the model's internal representations clearly distinguish correct spatial concepts, and Logit Lens (Belrose et al., 2023), a technique that projects intermediate hidden states directly

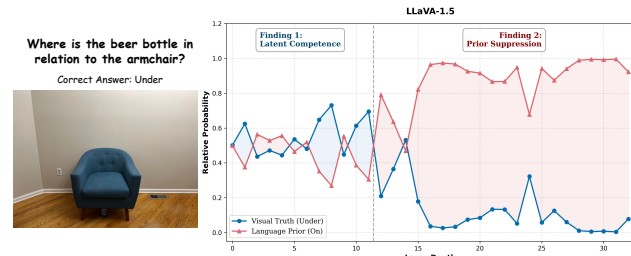

*Figure 2.* **Internal conflict analysis via Logit Lens.** This example illustrates the normalized relative probability between the visual truth ("under") and the language prior ("on"). By projecting hidden states to the vocabulary space, we observe that latent visual competence (Finding 1) is suppressed by strong language priors (Finding 2) during the late-stage decoding process.

onto the vocabulary space to reveal the model's instantaneous prediction at each layer. Specifically, we freeze all parameters of the LVLM and extract the hidden states of the last token from each decoder layer. For linear probing, we conduct two sets of experiments: **(1) In-distribution (ID) probing:** The *Controlled Images* dataset is randomly split into a training set (10%) and a test set (90%); **(2) Out-of-distribution (OOD) probing:** Probes are trained on the real-world *COCO_QA* dataset and evaluated on the clean-background *Controlled Images* dataset. Experimental results are illustrated in Figure 1 and Figure 2. Based on the contrast between the probing curves and the logit lens trajectories, we summarize two key findings.

### 3.1. Finding 1: Latent Geometric Competence

The internal analysis reveals a counter-intuitive phenomenon: LVLMs' intermediate layers have already accurately encoded spatial geometric information, with precision far exceeding the model's final output performance. As shown in Figure 1, for both LLaVA-1.5 and Qwen2.5-VL, the ID probing accuracy (blue line) rapidly ascends to near 100% in intermediate layers and maintains a high plateau. More critically, the OOD probing accuracy (orange line) which represents generalization capability also reaches a significant peak at specific layers (Layer 14 for LLaVA, Layer 28 for Qwen), substantially surpassing the grey baseline of final output accuracy. Corroborating this, the logit lens analysis (Figure 2, blue trajectory) demonstrates that the visually correct token ("under") frequently dominates the probability distribution in the early layers. This phenomenon effectively refutes the conventional wisdom that models lack spatial perception capabilities. It proves the existence of a high-confidence intrinsic geometric subspace within the geometric manifold constructed by positional encodings. Rather than being *blind* to spatial relationships, the model harbors correct visual signals that are latent within the representation manifold of intermediate layers, awaiting to be awakened by the correct mechanism.

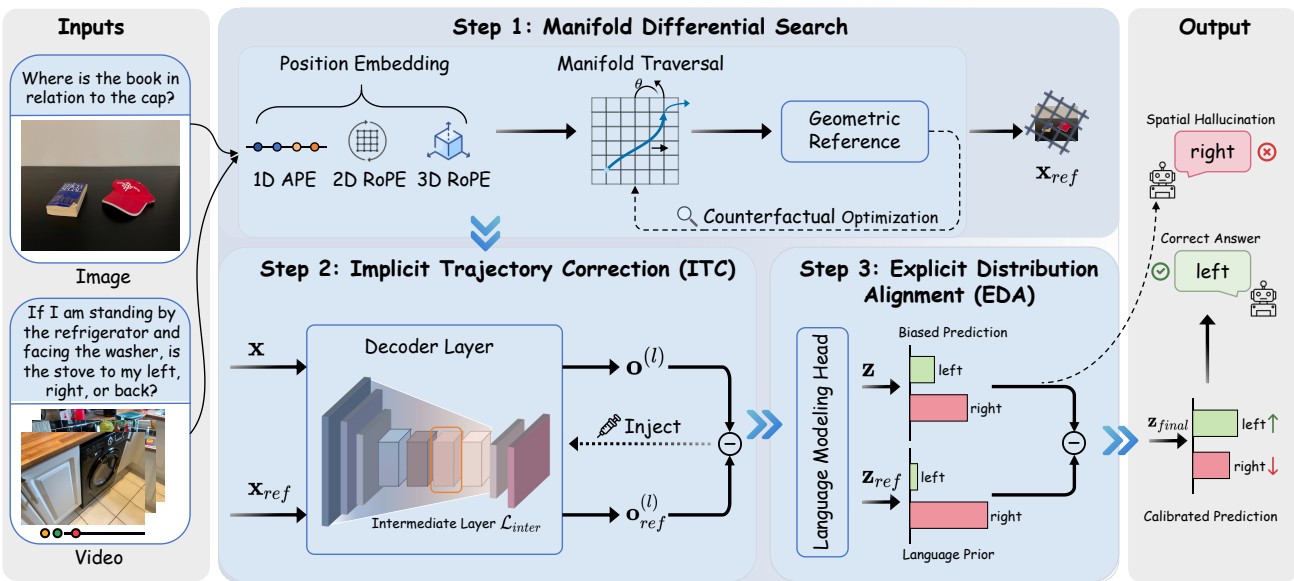

*Figure 3.* **Overview of GRASP.** We employ Manifold Differential Search to identify an optimal geometric counterfactual on the positional manifold. Guided by this anchor, Implicit Trajectory Correction (ITC) rectifies internal latent features to awaken spatial capabilities, while Explicit Distribution Alignment (EDA) reshapes the output distribution to counteract language prior dominance.

## 3.2. Finding 2: Language Prior Suppression

Although intermediate layers successfully encode spatial features, this competence suffers a severe representation-output misalignment during transmission to the output layer. Observing the OOD probing curves (orange line) in Figure 1, we notice that after peaking, the accuracy experiences a precipitous decline as the layer depth increases. This indicates that generic visual-geometric representations are overwhelmed by potent language priors during the deep decoding stages. To verify this mechanism at a granular level, we employ logit lens to visualize the token evolution of a typical halucinated sample in Figure 2. The trajectory reveals a distinct semantic collapse: while the visually correct token ("under") dominates the intermediate layers, a phase transition occurs in the deep layers where the hallucinated token driven by language priors ("on") surges, effectively overwriting the correct visual evidence. This shift from visual-semantic dominance to language-probability dominance directly causes a cognitive-expressive dissociation: the model intuitively *perceives* the correct orientation in its subconscious but is constrained by linguistic habits to *articulate* an erroneous answer.

## 4. Method

Addressing the findings in Section 3, we propose the **GRASP** framework. We model the spatial reasoning of LVLMs as a trajectory generation problem on a geometric manifold. As shown in Figure 3, GRASP employs manifold differential search to locate counterfactual references, which then drive a dual-level geometric rectification mechanism,

achieving training-free inference-time geometric manifold adaptation.

### 4.1. Manifold Differential Search

To shatter language confirmation bias, we construct a geometric counterfactual reference $\mathbf{x}_{ref}$ by strictly traversing the manifold structured by positional encodings. Unlike unconstrained perturbations that often corrupt semantics, this geometric constraint inherently preserves object identity, ensuring $\mathbf{x}_{ref}$ represents a valid spatial reconfiguration rather than arbitrary feature distortion.

**Traversing the Geometric Manifold.** Based on the definition in Section 2.1, the visual-spatial perception of LVLMs relies on positional encoding $\phi(\mathbf{c})$. From a Group Theory perspective, positional encoding fundamentally constructs a geometric manifold $\mathcal{M}$ endowed with a Lie Group Structure (Bronstein et al., 2021). Consequently, we define a parameterized geometric transformation operator $\mathcal{T}_\theta$, enabling smooth movement across the manifold by optimizing the parameter $\theta$. Let the original coordinates be $\mathbf{c} = [u, v]^T$; the transformed coordinates $\mathbf{c}'(\theta)$ are computed as:

$$\mathbf{c}'(\theta) = \begin{bmatrix} \cos\theta & -\sin\theta \\ \sin\theta & \cos\theta \end{bmatrix} \begin{bmatrix} u - c_x \\ v - c_y \end{bmatrix} + \begin{bmatrix} c_x \\ c_y \end{bmatrix}, \quad (1)$$

where $(c_x, c_y)$ denotes the image center. This transformation is essentially a group action of the special orthogonal group $SO(2)$ on the spatial coordinates, guaranteeing that generated samples remain strictly within the geometric manifold. We adapt $\mathcal{T}_\theta$ to three mainstream encodings, ensuring geometric topological equivariance: **(1) 1D APE:** Treating

$\mathbf{P}$ as sampling points on a discrete manifold, we perform bilinear interpolation resampling based on $\mathbf{c}'(\theta)$. **(2) 2D RoPE:** Leveraging rotation properties in the complex domain, we directly drive the phase shift of frequency bases on the manifold via $\mathbf{c}'(\theta)$. **(3) 3D RoPE:** For video data, we update only the spatial coordinates $\mathbf{c}'(\theta)$ within each frame while holding the temporal index $t$ invariant. This ensures $\mathbf{x}_{ref}$ undergoes a purely spatial geometric inversion while rigorously preserving original temporal logic and object appearance features.

**Searching for the Optimal Counterfactual.** We model the search for the optimal reference point $\theta^*$ as a traversal along the Geodesic on the manifold that maximizes semantic discriminability. We define the objective function $J(\theta)$ as the potential for spatial semantic discriminability. Our goal is to find an angle where the model exhibits high confidence in an counterfactual spatial relationship, thereby inducing maximal semantic conflict:

$$J(\theta) = P_{top1}(\mathbf{x}_{ref}; \theta) - P_{top2}(\mathbf{x}_{ref}; \theta), \qquad (2)$$

where $P_{topk}$ represents the probability of the $k$-th highest token. Maximizing this probability margin essentially locates a semantic pole on the manifold. Due to the high-dimensional non-convexity of the manifold, we employ Zeroth-Order Optimization to estimate gradients within the tangent space. Instead of sequential probing, we adopt a parallel sampling strategy. Specifically, we define a local sampling distribution $\mathcal{N}(\mu, \sigma^2 \mathbf{I})$ centered at the current state. We simultaneously sample a batch of tangent perturbations $\{\epsilon_i\}_{i=1}^{N}$ to generate a set of candidate geometric transformations $\Theta = \{\theta_i | \theta_i = \mu + \epsilon_i\}$. This enables the concurrent evaluation of the objective $J(\theta_i)$ across the entire candidate set. The optimal descent direction is then estimated by aggregating the contrastive signals via Monte Carlo Gradient Approximation to iteratively update the search parameters:

$$\nabla_\mu J \approx \frac{1}{N} \sum_{i=1}^{N} w_i \cdot \epsilon_i,$$
$$\text{where} \quad w_i = \frac{J(\mu + \epsilon_i) - \mathbb{E}[J]}{\sqrt{\text{Var}[J]} + \delta}. \qquad (3)$$

Here, $w_i$ represents the standardized importance weight derived from the objective values. This standardization acts as a variance reduction technique, ensuring robust gradient estimation on the manifold tangency. By parallelizing the candidate evaluation, this approach allows us to approximate the optimal counterfactual state $\theta^*$ through batched gradient estimation, effectively avoiding the local optima often associated with single-point iterative methods. The detailed pseudocode is provided in Appendix Algorithm 1.

### 4.2. Implicit Trajectory Correction

Upon acquiring the optimal reference $\mathbf{x}_{ref}(\theta^*)$, we execute implicit trajectory correction. This aims to distill pure geometric signals from latent representations and perform orthogonal manifold rectification on the collapsed reasoning trajectory. According to the analysis in Section 3.2, the distribution of spatial geometric information evolves dynamically, reaching its optimal alignment within the intermediate layers $\mathcal{L}_{inter}$. Therefore, we define the geometric signal $\mathbf{v}^{(l)}$ as the normalized difference between the self-attention outputs $\mathbf{o}^{(l)}$ and $\mathbf{o}_{ref}^{(l)}$:

$$\mathbf{v}^{(l)} = \frac{\mathbf{o}^{(l)} - \mathbf{o}_{ref}^{(l)}}{\|\mathbf{o}^{(l)} - \mathbf{o}_{ref}^{(l)}\|_2}. \qquad (4)$$

This differential calculation cancels out rotation-invariant non-spatial semantics (e.g., texture, color), retaining only the manifold tangent direction sensitive to coordinate transformations. We target the self-attention module rather than the feed-forward network (FFN) because, as outlined in Section 2.1, the former acts as the geometric operator for constructing spatial topology while the latter prioritizes the processing of factual knowledge, and stripping FFN signals thus minimizes semantic noise interference. Subsequently, we apply the intervention across the identified intermediate window $\mathcal{L}_{inter}$:

$$\tilde{\mathbf{o}}^{(l)} = \mathbf{o}^{(l)} + \alpha \cdot \mathbf{v}^{(l)}, \quad \forall l \in \mathcal{L}_{inter}, \qquad (5)$$

where $\alpha$ is the correction strength. This mechanism functions as a geometric restorative force, pulling the deviating latent representation back onto the manifold surface with the correct geometric curvature, thereby compelling the model to maintain spatial structural perception throughout deep decoding.

### 4.3. Explicit Distribution Alignment

Although implicit correction safeguards internal features, Finding 3.2 suggests that language priors at the output layer may still distort the final generation distribution. To counteract this, we introduce Explicit Distribution Alignment, which leverages the reference sample $\mathbf{x}_{ref}$ as a counterfactual anchor to reshape the prediction distribution. Let $\mathbf{z}$ and $\mathbf{z}_{ref}$ denote the logits for the original and reference samples, respectively. The aligned distribution $\mathbf{z}_{final}$ is defined as:

$$\mathbf{z}_{final} = \mathbf{z} + e^{-d} \cdot (\mathbf{z} - \mathbf{z}_{ref}), \qquad (6)$$

where $d = \text{JSD}(\text{Softmax}(\mathbf{z}) \parallel \text{Softmax}(\mathbf{z}_{ref}))$ quantifies the model's sensitivity to geometric transformations. This dynamic weight $e^{-d}$ acts as an adaptive gate: when $d$ is small, indicating the model ignores geometric changes due to strong priors, $e^{-d}$ approaches 1, fully activating the correction; conversely, a large $d$ implies high sensitivity, naturally decaying the weight to preserve valid predictions.

*Table 1.* **Main Results on WhatsUp.** We report performance across *Controlled*, *COCO*, and *VG* subsets (Metrics in $\times 10^{-2}$). Best-performing methods per model are highlighted in **bold**. P Acc and S Acc denote Pair Accuracy and Set Accuracy, respectively.

| Model | Controlled_A | | | Controlled_B | | | COCO_one | COCO_two | VG_one | VG_two |
|---|---|---|---|---|---|---|---|---|---|---|
| | Acc | P Acc | S Acc | Acc | P Acc | S Acc | Acc | Acc | Acc | Acc |
| LLaVA-1.5 | 60.3 | 40.6 | 0.0 | 73.1 | 41.6 | 3.7 | 53.0 | 58.2 | 35.9 | 40.8 |
| +DoLa | 61.2 ↑0.9 | 41.6 ↑1.0 | 0.0 | 73.4 ↑0.3 | 42.2 ↑0.6 | 3.7 | 53.7 ↑0.7 | 57.5 ↓0.7 | 36.2 ↑0.3 | 42.1 ↑1.3 |
| +VCD | 61.5 ↑1.2 | 39.4 ↓1.2 | 0.0 | 73.4 ↑0.3 | 42.2 ↑0.6 | 3.7 | 53.3 ↑0.3 | 58.2 | 35.8 ↓0.1 | 42.5 ↑1.7 |
| +AdaptVis | 84.9 ↑24.6 | 61.2 ↑20.6 | 30.3 ↑30.3 | 83.8 ↑10.7 | 55.7 ↑14.1 | 18.3 ↑14.6 | 53.6 ↑0.6 | 59.9 ↑1.7 | 42.7 ↑6.8 | 48.1 ↑7.3 |
| **+GRASP** | **86.4** ↑26.1 | **61.2** ↑20.6 | **32.7** ↑32.7 | 83.5 ↑10.4 | **55.7** ↑14.1 | **19.6** ↑15.9 | **61.4** ↑8.4 | **62.8** ↑4.6 | **48.5** ↑12.6 | **57.5** ↑16.7 |
| Qwen2.5-VL | 90.0 | 63.0 | 32.7 | 93.6 | 70.3 | 41.6 | 68.6 | 71.0 | 65.7 | 70.4 |
| +DoLa | 90.3 ↑0.3 | 63.0 | 31.5 ↓1.2 | 93.6 | 70.3 | 41.6 | 69.7 ↑1.1 | 73.3 ↑2.3 | 67.9 ↑2.2 | 71.7 ↑1.3 |
| +VCD | 92.1 ↑2.1 | 67.1 ↑4.1 | 39.2 ↑6.5 | 93.6 | 70.3 | 41.6 | 68.7 ↑0.1 | 71.6 ↑0.6 | 65.4 ↓0.3 | 69.1 ↓1.3 |
| +AdaptVis | 90.0 | 63.0 | 32.7 | 93.7 ↑0.1 | 70.5 ↑0.2 | 41.7 ↑0.1 | 68.7 ↑0.1 | 71.2 ↑0.2 | 66.3 ↑0.6 | 71.7 ↑1.3 |
| **+GRASP** | **97.6** ↑7.6 | **75.8** ↑12.8 | **52.1** ↑19.4 | **93.9** ↑0.3 | **70.9** ↑0.6 | **42.8** ↑1.2 | **70.6** ↑2.0 | **74.7** ↑3.7 | **72.4** ↑6.7 | **75.1** ↑4.7 |

## 5. Experiments

### 5.1. Experiment Setup

**Benchmarks.** We evaluate image-based spatial reasoning on the widely used WhatsUp (Kamath et al., 2023) and VSR (Liu et al., 2023) benchmarks. WhatsUp comprises synthetic and realistic domains: the synthetic data is divided into *Controlled_A* (large-small object pairs) and *Controlled_B* (two small objects) , while the realistic data from *MS COCO* (Lin et al., 2014) and *Visual Genome (VG)* (Krishna et al., 2017) is categorized into one-object and two-object subsets. Following AdaptVis (Chen et al., 2025b), we reformat these into generative multiple-choice QA tasks, applying a four-option setting ⟨ *left, right, on, under* ⟩ for *Controlled* and *COCO* subsets, and extending *VG* to a six-option setting ⟨ *left, right, on, under, front, behind* ⟩. For VSR, we leverage QA formats generated by GPT-4o (Hurst et al., 2024) to assess the model's generative spatial understanding capabilities. Furthermore, for video-based spatial reasoning, we select the *Relative Direction* subtask from VSI-Bench (Yang et al., 2025b) to evaluate the model's ability to accurately determine relative object positions amidst temporal dynamics.

**Evaluation Metrics.** We report accuracy of exact match across all benchmarks. For VSR, we additionally report the F1 score. Furthermore, leveraging the unique contrastive attributes of the *Controlled Images* subset in WhatsUp, we introduce consistency metrics to eliminate random guessing: Pair Accuracy requires the model to correctly answer both queries within a mirrored spatial relationship pair, while Set Accuracy mandates correct predictions for all spatial relations within a specific scene.

**Baselines.** We compare GRASP with representative inference-time interventions, including contrastive decoding strategies DoLa (Chuang et al., 2023), VCD (Leng et al., 2024) and attention intervention methods AdaptVis (Chen et al., 2025b). All settings of baseline methods follow the

*Table 2.* **Generalization on video.** Accuracy on the *Relative Direction* subset of VSI-Bench. We compare GRASP and its ablation components, Implicit Trajectory Correction (ITC) and Explicit Distribution Alignment (EDA), against vanilla baselines.

| Method | Relative Direction | | |
|---|---|---|---|
| | Easy | Medium | Hard |
| LLaVA-NeXT-Video | 52.1 | 45.2 | 31.1 |
| + ITC | 54.4 ↑2.3 | 48.4 ↑3.2 | 37.8 ↑6.7 |
| + EDA | 52.3 ↑0.2 | 46.6 ↑1.4 | 33.0 ↑1.9 |
| **+ GRASP (Full)** | **54.9** ↑2.8 | **48.7** ↑3.5 | **39.0** ↑7.9 |
| Qwen3-VL | 48.4 | 49.2 | 35.9 |
| + ITC | 55.3 ↑6.9 | 52.2 ↑3.0 | 45.0 ↑9.1 |
| + EDA | 49.3 ↑0.9 | 50.7 ↑1.5 | 37.0 ↑1.1 |
| **+ GRASP (Full)** | **56.2** ↑7.8 | **52.4** ↑3.2 | **45.6** ↑9.7 |

default configurations from the original papers.

**Implementation Details.** We verify the versatility of GRASP across four representative LVLMs, ensuring coverage of distinct positional encoding paradigms across both image and video modalities: **(1) 1D APE Models:** LLaVA-1.5 7B (Liu et al., 2024a) (Image) and LLaVA-NeXT-Video-7B (Liu et al., 2024b) (Video); **(2) RoPE Models:** Qwen2.5-VL 3B (Bai et al., 2025) (Image, 2D RoPE) and Qwen3-VL 8B (Yang et al., 2025a) (Video, 3D RoPE). Greedy decoding is adopted as the default inference strategy. All experiments were conducted on NVIDIA A100 (40 GB) GPUs using the PyTorch framework. Comprehensive hyperparameter settings are detailed in Appendix E.2.

### 5.2. Main Results

**Results on Image Benchmarks.** We focus our primary analysis on the WhatsUp benchmark (Table 1), deferring detailed VSR results to Appendix F.1 due to space constraints. GRASP consistently outperforms baseline methods across diverse domains. Notably, on LLaVA-1.5 (1D APE), GRASP achieves a peak accuracy gain of 26.1% on the

*Table 3.* **Component Ablation.** We validate the contribution of Implicit Trajectory Correction (ITC) and Explicit Distribution Alignment (EDA). The full GRASP framework consistently outperforms individual modules.

| Method | Controlled_A | | |
|---|---|---|---|
| | Acc | P Acc | S Acc |
| LLaVA-1.5 | 60.3 | 40.6 | 0.0 |
| + ITC | 83.9 | 60.0 | 27.9 |
| + EDA | 73.3 | 49.1 | 14.5 |
| **+ GRASP (Full)** | **86.4** | **61.2** | **32.7** |
| Qwen2.5-VL | 90.0 | 63.0 | 32.7 |
| + ITC | 93.3 | 68.5 | 41.2 |
| + EDA | 95.8 | 73.3 | 49.7 |
| **+ GRASP (Full)** | **97.6** | **75.8** | **52.1** |

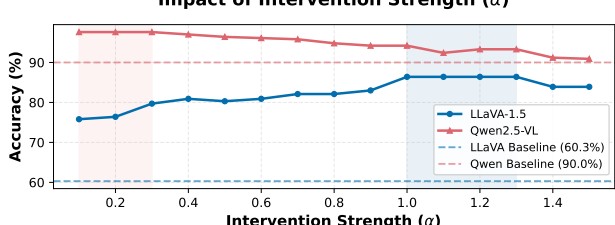

*Figure 4.* **Impact of Intervention Strength ($\alpha$).** GRASP consistently surpasses baselines (dashed lines). Shaded regions highlight optimal $\alpha$ ranges. LLaVA requires over-rectification ($\alpha >= 1.0$) to counter deep language priors, whereas Qwen is awakened by a subtle nudge ($\alpha \approx 0.2$), reflecting lower calibration resistance.

*Controlled_A* subset and increases the Set Accuracy from 0.0% to 32.7%. This dramatic boost in set-level consistency validates the effectiveness of implicit trajectory correction in locking the intrinsic geometric truth against volatile language priors. Compared to AdaptVis, GRASP demonstrates superior robustness in realistic scenarios, achieving a 16.7% gain on *VG_two*. Furthermore, GRASP proves effective across different positional encoding paradigms: on the stronger Qwen2.5-VL backbone (2D RoPE), it further boosts performance on the *Controlled_A* subset by 7.6% to reach 97.6% accuracy, confirming the universality of Manifold Differential Search across discrete and continuous manifolds. In contrast, general hallucination mitigation methods like DoLa and VCD fail to provide meaningful improvements, as they primarily target language uncertainties while neglecting the root cause of geometric misalignment.

**Generalization on Video Benchmarks.** We evaluate GRASP on the *Relative Direction* subset of VSI-Bench, which categorizes samples into *Easy*, *Medium*, and *Hard* levels. Given the absence of existing training-free interventions for video tasks, we benchmark directly against vanilla base models. Table 2 reports the performance of GRASP and its ablation modules. Results confirm that GRASP effectively generalizes to spatiotemporal manifolds, consistently improving both 1D APE (LLaVA-NeXT-Video) and 3D RoPE (Qwen3-VL) architectures. Notably, the full framework yields the most robust gains, achieving significant improvements of 7.9% and 9.7% on the challenging *Hard* subset for LLaVA and Qwen, respectively.

### 5.3. Ablation Studies

We conduct all ablation studies on the *Controlled_A* subset, utilizing LLaVA-1.5 and Qwen2.5-VL to evaluate the framework across varying model capacities.

**Component Effectiveness.** Table 3 decomposes the contribution of each module within our dual-level rectification framework. **(1) Implicit Trajectory Correction (ITC)**

awakens latent competence. For LLaVA-1.5, ITC alone yields a 23.6% accuracy gain, confirming the existence of a suppressed geometric subspace that can be distilled to rectify the inference trajectory. **(2) Explicit Distribution Alignment (EDA) bridges the representation-output gap.** On the stronger Qwen2.5-VL backbone, EDA outperforms ITC, suggesting the primary bottleneck shifts to misalignment; EDA effectively breaks this barrier by penalizing output-stage language dominance. **(3) Synergy of GRASP.** The full framework yields the best performance, demonstrating that sustaining awakened signals requires simultaneous intervention on both the trajectory path and the output exit.

**Impact of Intervention Strength ($\alpha$).** Figure 4 illustrates the sensitivity to intervention strength $\alpha$. The optimal $\alpha$ varies significantly across architectures, reflecting distinct internal mechanisms of spatial hallucination. LLaVA-1.5 demands over-rectification peaking at $\alpha \in [1.0, 1.3]$, to escape the deep energy well of its language priors. In contrast, Qwen2.5-VL exhibits low calibration resistance, peaking early at $\alpha \in [0.1, 0.3]$; a subtle nudge suffices to awaken latent capabilities, while excessive intervention disrupts alignment. Accordingly, we adopt $\alpha = 1.0$ for LLaVA-1.5 and $\alpha = 0.3$ for Qwen2.5-VL as default settings. Crucially, GRASP consistently outperforms baselines across the full $\alpha$ spectrum, confirming its universal robustness.

### 5.4. Analysis

**Recovery of Latent Geometric Signals.** To verify whether GRASP genuinely awakens latent geometric capabilities, we conducted OOD linear probing on the intervened model. As shown in Figure 5, the rectified OOD probe demonstrates that intervention within the manifold injection window effectively reverses semantic collapse. Crucially, in deep layers where the vanilla model suffered severe suppression, the rectified representations maintain high discriminability. This confirms that GRASP sustains these awakened spatial signals throughout decoding, ensuring the final output is grounded in visual reality.

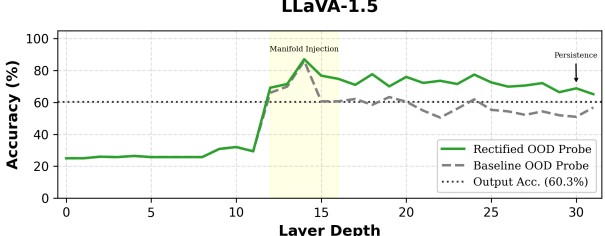

*Figure 5*. **Verification of Signal Awakening.** The Rectified OOD Probe demonstrates that Manifold Injection effectively reverses the vanilla model's semantic collapse. Crucially, the rectified trajectory in deep layers directly confirms that awakened spatial signals are sustained against language suppression throughout decoding.

*Table 4*. **Inference Efficiency.** Latency (ms/token ↓) and Throughput (tokens/s ↑) are compared. Values in parentheses denote relative cost normalized to Greedy decoding.

| Metric | Greedy | VCD | AdaptVis | GRASP |
|---|---|---|---|---|
| Latency | 37.60 | 76.70 (×2.04) | 78.89 (×2.09) | **64.28** (×1.71) |
| Throughput | 26.60 | 13.04 (×0.49) | 12.68 (×0.48) | **15.56** (×0.58) |

**Inference-Time Efficiency.** Table 4 demonstrates GRASP's superior efficiency. The key advantage lies in avoiding complete autoregressive re-generation. Unlike baselines method that often require generating full sequences twice or maintaining dual-stream decoding, GRASP only necessitates additional forward passes to compute the geometric rectification signal. While this incurs a moderate $1.71\times$ latency overhead compared to vanilla greedy decoding, it represents a highly cost-effective trade-off, securing substantial performance gains at a significantly lower computational cost than existing interventions.

# 6. Related work

### 6.1. Spatial Reasoning in LVLMs

Existing approaches to enhancing spatial reasoning in LVLMs generally fall into two paradigms: explicit spatial modeling (Cheng et al., 2024; Feng, 2025; Fan et al., 2025; Zhang et al., 2025a) and reasoning reinforcement (Zhao et al., 2025; Wu et al., 2025a). Explicit spatial modeling augments the perception of relative object positions by integrating external geometric priors. SpatialVLM (Chen et al., 2024) and SpatialBot (Cai et al., 2025) enrich the spatial structural information within visual representations by introducing geometric modalities; LocVLM (Ranasinghe et al., 2024) explicitly aligns visual features with spatial semantics via coordinate-based instruction tuning. Conversely, reasoning reinforcement methods attempt to elicit spatial intelligence using Chain-of-Thought (CoT) prompting or reinforcement learning (RL). Notably, SpaceR (Ouyang et al., 2025) and Video-R1 (Feng et al., 2025) have achieved sig-

nificant progress in spatial relationship reasoning through GRPO-based RL. However, Liao et al. (2025) show that without costly RL, standard CoT often fails to activate spatial perception. This suggests that spatial hallucinations stem not from a lack of geometric representation, but from its suppression by language priors. While recent studies have diagnosed similar suppression phenomena in broader visual contexts (Fu et al., 2025), resolving it in spatial reasoning necessitates direct inference-time interventions beyond mere logical deduction.

### 6.2. Inference-time Intervention Mechanisms

To avoid the high costs of retraining, recent research has pivoted towards inference-time intervention (Liu et al., 2025; Chen et al., 2025a; Wu et al., 2025b; Li et al., 2025b), primarily focusing on contrastive decoding (Wang et al., 2024a;b) and attention intervention strategies (Zhang et al., 2024; Kang et al., 2025). DoLa (Chuang et al., 2023) and VCD (Leng et al., 2024) aim to amplify factual signals and suppress hallucinations by contrasting logits from different decoder layers or distinct views. However, these methods primarily target language priors or object hallucinations, lacking specific modeling for spatial semantics. Alternatively, attention intervention methods (Liu et al., 2024c; Chen et al., 2025b) attempt to adjust internal attention distributions during inference to highlight key regions and suppress irrelevant information. Nevertheless, these approaches rely heavily on high-level semantic features, failing to directly perceive or utilize low-level geometric features. Building upon the insight that visual positional encoding acts as the decisive carrier of spatial semantics (Zhang et al., 2025b), we are the first to leverage the geometric equivariance of the positional encoding manifold to directly intervene in the model's coordinate perception system, effectively resolving spatial hallucinations.

# 7. Conclusion

In this work, we revisited spatial hallucinations in LVLMs from the perspective of mechanistic interpretability, revealing that they stem from a representation-output misalignment rather than perceptual deficits: latent geometric competence in intermediate decoder layers is suppressed by strong language priors. To address this, we introduce GRASP, a training-free inference-time intervention. By performing manifold differential search on the positional encoding manifold, GRASP leverages geometric counterfactual signals to drive Implicit Trajectory Correction and Explicit Distribution Alignment, rectifying suppressed geometric signals without parameter updates. Extensive experiments confirm GRASP's superior universality and robustness across diverse positional encoding manifolds and multimodal tasks. This paper not only validates the feasibility of awakening

latent geometric capabilities through inference-time adaptation but also paves a cost-effective path towards physically grounded vision-language models.

## Acknowledgements

The authors gratefully acknowledge the financial support provided by the Key Program of the National Natural Science Foundation of China (Grant No. 62436004) and the National Natural Science Foundation of China (Grant No. 62506253, Grant No. 62372317).

## Impact Statement

This paper presents work whose goal is to advance the field of Machine Learning, specifically by enhancing the spatial intelligence and geometric reasoning capabilities of Large Vision-Language Model. There are many potential societal consequences of our work, none which we feel must be specifically highlighted here.

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

## A. Limitations

**Dependence on Manual Hyperparameters.** A primary limitation of GRASP is its reliance on the manual specification of intervention hyperparameters, specifically the intervention strength ($\alpha$) and the selection of intervention layers. Currently, these parameters are determined based on empirical observations from the validation set. While our ablation studies indicate that GRASP is robust, consistently outperforming the vanilla model across a wide range of $\alpha$ values, achieving peak performance still necessitates architecture-specific tuning to align with the model's inherent calibration resistance.

**Parametric Constraints of the Geometric Manifold.** The geometric manifold defined in this work is currently parameterized by affine transformations. While highly effective for relative orientation, this simplified parametrization may be insufficient for scenes involving non-rigid deformations or complex 3D occlusions. Future research could explore incorporating physics-based 3D synthesis priors to construct higher-fidelity manifolds, further enhancing the physical consistency of the generated counterfactuals.

## B. Mathematical Derivations and Formal Proofs

In this section, we provide the detailed mathematical foundations supporting the **Inference-time Geometric Manifold Adaptation** paradigm proposed in the main text. We first formalize the topological structure of the positional manifolds constructed by APE and RoPE, then derive the exact formulations of the geometric operators used in GRASP, and finally provide the theoretical justification for the Manifold Differential Search algorithm.

### B.1. Topological Analysis of Positional Manifolds

In Section 2.2, we posit that positional encodings construct a Geometric Manifold $\mathcal{M}$ embedded in the feature space $\mathbb{R}^D$. Here, we formally analyze the structure of this manifold for both Absolute Positional Encoding (APE) and Rotary Positional Embedding (RoPE).

#### B.1.1. ROTARY POSITIONAL EMBEDDING (ROPE) AS A LIE GROUP MANIFOLD

**Proposition 1.** The operation of RoPE defines a continuous geometric manifold with a Lie Group structure isomorphic to the direct product of $SO(2)$ groups.

**Proof.** Recall that for a position index $m \in \mathbb{N}$ and a feature vector $\mathbf{x} \in \mathbb{R}^d$, RoPE applies a rotation matrix $\mathcal{R}_{m,\Theta}^d$. For a standard implementation where $d$ is even, the feature space is divided into $d/2$ subspaces. The rotation matrix $\mathcal{R}_{m,\Theta}^d$ is a block-diagonal matrix:

$$\mathcal{R}_{m,\Theta}^d = \begin{pmatrix} \mathbf{R}_1 & 0 & \cdots & 0 \\ 0 & \mathbf{R}_2 & \cdots & 0 \\ \vdots & \vdots & \ddots & \vdots \\ 0 & 0 & \cdots & \mathbf{R}_{d/2} \end{pmatrix}, \tag{7}$$

where each block $\mathbf{R}_j$ corresponds to the $j$-th frequency band $\omega_j = \theta_{base}^{-2(j-1)/d}$ and is defined as:

$$\mathbf{R}_j = \begin{pmatrix} \cos(m\omega_j) & -\sin(m\omega_j) \\ \sin(m\omega_j) & \cos(m\omega_j) \end{pmatrix}. \tag{8}$$

This matrix $\mathbf{R}_j$ is an element of the Special Orthogonal Group $SO(2)$, which represents all rotations in 2D Euclidean space.

Since the block diagonal matrix is formed by $SO(2)$ components, the entire transformation $\mathcal{R}_{m,\Theta}^d$ resides in the product group $G = SO(2) \times \cdots \times SO(2)$ ($d/2$ times). A key property of Lie Groups is Geometric Equivariance. Let the geometric transformation $\mathcal{T}_\delta$ be a translation of the coordinate $m$ by $\delta$. The transformed embedding is:

$$f(\mathbf{x}, m + \delta) = \mathcal{R}_{m+\delta,\Theta}^d \mathbf{x}. \tag{9}$$

Due to the homomorphism property of exponential maps in Lie Groups, $\mathcal{R}_{m+\delta} = \mathcal{R}_\delta \cdot \mathcal{R}_m$. Thus:

$$f(\mathbf{x}, m + \delta) = \mathcal{R}_{\delta,\Theta}^d (\mathcal{R}_{m,\Theta}^d \mathbf{x}). \tag{10}$$

This proves that moving on the coordinate manifold (changing $m$ to $m + \delta$) is mathematically equivalent to applying a group action (rotation $\mathcal{R}_\delta$) on the feature manifold. This continuous differentiability with respect to position $m$ allows us to define continuous gradients on the manifold, justifying our Manifold Differential Search.

B.1.2. ABSOLUTE POSITIONAL ENCODING (APE) AS A DISCRETIZED MANIFOLD

**Definition 1 (Discretized Manifold).** We define the APE parameter matrix $\mathbf{P} \in \mathbb{R}^{H \times W \times D}$ as a discrete sampling of a continuous function $f : \mathcal{C} \to \mathbb{R}^D$, where $\mathcal{C} \subset \mathbb{R}^2$ is the continuous coordinate space.

In standard Vision Transformers (e.g., ViT, LLaVA), $\mathbf{P}$ is learnable. However, to perform geometric transformations (like rotation) that map integer coordinates $(h, w)$ to non-integer coordinates $(h', w')$, we must define the continuous extension of this manifold.

We adopt the Bilinear Manifold Interpolation. Let $\mathbf{c}' = (u, v)$ be a continuous coordinate after transformation. The embedding vector $\mathbf{p}(\mathbf{c}')$ is computed by projecting $\mathbf{c}'$ onto the grid formed by the four nearest discrete neighbors: top-left $\mathbf{c}_{tl} = (\lfloor u \rfloor, \lfloor v \rfloor)$, top-right $\mathbf{c}_{tr}$, bottom-left $\mathbf{c}_{bl}$, and bottom-right $\mathbf{c}_{br}$.

The embedding on the continuous manifold is defined as:

$$\mathbf{p}(u, v) = \sum_{k \in \{tl, tr, bl, br\}} w_k(u, v) \cdot \mathbf{P}[\mathbf{c}_k], \tag{11}$$

where the weights $w_k$ are the areas of the opposing rectangles, e.g., $w_{tl} = (\lceil u \rceil - u)(\lceil v \rceil - v)$. This interpolation ensures that the function $f(\mathbf{c})$ is $C^0$ continuous everywhere and differentiable almost everywhere (except at integer grid lines), allowing for gradient estimation via zero-order optimization.

## B.2. Derivation of Geometric Operators

In GRASP, we define the operator $\mathcal{T}_\theta$ to traverse the manifold. Here we derive the exact transformation matrices for 2D spatial data (Images) and 3D spatiotemporal data (Videos).

B.2.1. THE 2D SPATIAL OPERATOR (IMAGES)

For an image with height $H$ and width $W$, we define the center of rotation as $\mathbf{c}_0 = (c_x, c_y) = (W/2, H/2)$. Let the original coordinate be $\mathbf{c} = [x, y]^T$. The rotation operator $\mathcal{T}_\theta^{2D}$ rotates the coordinate system by an angle $\theta$ around $\mathbf{c}_0$. The transformed coordinate $\mathbf{c}' = [x', y']^T$ is derived using the standard affine rotation matrix:

$$\begin{aligned} \begin{bmatrix} x' \\ y' \end{bmatrix} &= \mathcal{T}_\theta^{2D}(\mathbf{c}) \\ &= \begin{bmatrix} \cos\theta & -\sin\theta \\ \sin\theta & \cos\theta \end{bmatrix} \begin{bmatrix} x - c_x \\ y - c_y \end{bmatrix} + \begin{bmatrix} c_x \\ c_y \end{bmatrix}. \end{aligned} \tag{12}$$

**Boundary Handling:** Since the rotated coordinates $\mathbf{c}'$ might fall outside the original image domain $[0, W] \times [0, H]$, we apply a Reflection Padding strategy on the manifold. Mathematically, for a dimension with limit $L$, if a coordinate $u' \notin [0, L]$, we map it to $L - |u' - L|$ (if $u' > L$) or $|u'|$ (if $u' < 0$). This ensures the query always stays within the defined embedding lookup table.

B.2.2. THE 3D SPATIOTEMPORAL OPERATOR (VIDEOS)

For video inputs, the coordinate system is 3-dimensional: $\mathbf{c} = (t, h, w)$, where $t$ is the temporal frame index.

A critical design choice in GRASP is the Temporal Preservation Constraint. Since our goal is to rectify spatial hallucinations (e.g., "left/right") without disrupting the causal logic of the video (e.g., "action A happens before action B"), the geometric operator must act strictly on the spatial subspace.

We define the 3D operator $\mathcal{T}_\theta^{3D}$ as a block matrix acting on the augmented coordinate vector $[t, h, w, 1]^T$:

$$\mathcal{T}_\theta^{3D} = \begin{pmatrix} 1 & 0 & 0 & 0 \\ 0 & \cos\theta & -\sin\theta & \Delta_h(\theta) \\ 0 & \sin\theta & \cos\theta & \Delta_w(\theta) \\ 0 & 0 & 0 & 1 \end{pmatrix}, \tag{13}$$

where $\Delta_h(\theta) = c_h(1 - \cos\theta) + c_w \sin\theta$ and $\Delta_w(\theta) = c_w(1 - \cos\theta) - c_h \sin\theta$ handle the translation back to the center. This ensures that $t' = t$, while $(h, w)$ undergoes spatial rotation. Consequently, for 3D RoPE, the temporal frequency components remain unchanged, preserving the video's temporal coherence.

### B.3. Theoretical Justification for Parallel Gradient Estimation

The core of GRASP involves maximizing the semantic conflict objective $J(\theta)$ using a **Stochastic Zeroth-Order Optimization** strategy. Here we provide the theoretical basis for why this batched gradient estimation yields a robust geometric signal without sequential iteration.

#### B.3.1. GRADIENT ESTIMATION VIA GAUSSIAN SMOOTHING

Since the objective function $J(\theta)$ involves the full forward pass of the LVLM and discrete token selection, it is non-differentiable or possesses shattered gradients. We approximate $J(\theta)$ with a Gaussian-smoothed function $J_\sigma(\theta)$:

$$J_\sigma(\theta) = \mathbb{E}_{\epsilon \sim \mathcal{N}(0,\mathbf{I})}[J(\theta + \sigma\epsilon)], \tag{14}$$

where $\sigma$ is a smoothing parameter. According to Nesterov's Theorem on Random Gradient-Free Minimization, the gradient of this smoothed function is well-defined and can be computed as:

$$\nabla J_\sigma(\theta) = \mathbb{E}_{\epsilon \sim \mathcal{N}(0,\mathbf{I})}\left[\frac{J(\theta + \sigma\epsilon) - J(\theta)}{\sigma}\epsilon\right]. \tag{15}$$

This theoretical guarantee implies that the optimal descent direction exists and is computable via random sampling even in the absence of analytical derivatives.

#### B.3.2. DERIVATION OF MONTE CARLO APPROXIMATION

In our parallel implementation (Algorithm 1), we employ a **Monte Carlo approximation** of the expectation in Eq. 3 using concurrent sampling. Instead of calculating a temporal update trajectory, we estimate the global gradient direction $\mathbf{d}^*$ directly from the simultaneous batch $\{\epsilon_i\}_{i=1}^N$. We formulate this as a standard variance-reduced gradient estimator. By sampling $N$ directions in the tangent space, we approximate the gradient of the expected objective. Using the standardized importance weight $w_i$ defined in the main text, the estimator for the optimal manifold direction becomes:

$$\hat{\nabla}J(\theta) \approx \frac{1}{N\sigma}\sum_{i=1}^N w_i \cdot \epsilon_i. \tag{16}$$

Here, $w_i$ acts as a control variate that re-weights the contribution of each tangent vector based on its standardized deviation from the mean, minimizing the estimation variance.

#### B.3.3. CONVERGENCE ANALYSIS OF PARALLEL ESTIMATION

The validity of replacing sequential iterations with a single parallel batch relies on the Law of Large Numbers. As the sample size $N$ increases, the Monte Carlo estimator converges to the true smoothed gradient:

$$\lim_{N\to\infty} \frac{1}{N}\sum_{i=1}^N w_i \cdot \epsilon_i \to \mathbb{E}_\epsilon[w(\epsilon)\epsilon] \propto \nabla J_\sigma(\theta). \tag{17}$$

Consequently, our **Parallel Monte Carlo Search** effectively performs a one-step gradient descent in the parameter space. By leveraging a batch size $N$, we achieve a high-fidelity estimation of the geometric counterfactual reference direction in a single forward pass, thereby avoiding the latency accumulation of sequential steps while maintaining the theoretical rigor of gradient-based optimization.

## C. Algorithm Implementation Details

In this section, we detail the procedural workflow of Manifold Differential Search. To reconcile the requirement for rigorous optimal counterfactuals with the efficiency demands of inference-time adaptation, we formulate the optimization process as a parallelized global search.

Since the objective function $J(\theta)$ is non-differentiable with respect to the geometric parameter $\theta$, we employ a Zeroth-Order Gradient Estimation strategy. Unlike sequential iterative methods that are sensitive to initialization, our approach constructs

a concurrent batch of geometric transformations. By simultaneously evaluating multiple tangent perturbations, we estimate the robust descent direction in a single forward pass, effectively avoiding local optima while minimizing temporal latency.

The detailed algorithm is presented in Algorithm 1.

---

**Algorithm 1** Manifold Differential Search

1: **Input:** Visual Input $\mathbf{x}$, Query $q$, Manifold Operator $\mathcal{T}_\theta$, Encoder $\mathcal{V}$
2: **Hyperparameters:** Sampling Candidates $N$, Search Radius $\sigma$, Learning Rate $\eta$, Iterations $T$
3: **Output:** Optimal Geometric Signal $\mathbf{v}^*$
4: Initialize search distribution center $\mu \leftarrow \theta_{init}$
5: Initialize global best: $J^* \leftarrow -\infty, \mathbf{v}^* \leftarrow$ None
6: **for** $t = 1$ to $T$ **do**
7:     {**Phase 1: Parallel Tangent Sampling**}
8:     **Simultaneously sample** tangent perturbations:
9:         $\mathcal{E} = \{\epsilon_i\}_{i=1}^N \sim \mathcal{N}(0, \sigma^2 \mathbf{I})$
10:     Generate Candidate Transformations:
11:         $\Theta = \{\theta_i \mid \theta_i = \mu + \epsilon_i, \forall \epsilon_i \in \mathcal{E}\}$
12:     {**Phase 2: Batched Manifold Evaluation**}
13:     **Parallel** Feature Extraction & Vector Computation:
14:         $\mathbf{X}_{batch} \leftarrow \text{Concat}([\mathcal{T}_{\theta_i}(\mathbf{x}) \text{ for } \theta_i \in \Theta])$
15:         $\mathbf{V}_{batch} \leftarrow \text{Normalize}(\mathcal{V}(\mathbf{x}) - \mathcal{V}(\mathbf{X}_{batch}))$
16:     **Batched** Trial Intervention & Reward Calculation:
17:         $\mathbf{Z}_{batch} \leftarrow \text{Decoder}(\mathbf{x}; \text{inject} = \mathbf{V}_{batch})$
18:         $\mathcal{J} = \{J(\theta_i) \leftarrow \text{Reward}(\mathbf{Z}_{batch}[i])\}_{i=1}^N$
19:     {**Phase 3: Monte Carlo Gradient Estimation**}
20:     Track best candidate:
21:         $k = \arg\max \mathcal{J}; \textbf{if } \mathcal{J}[k] > J^* \textbf{ then } J^* \leftarrow \mathcal{J}[k], \mathbf{v}^* \leftarrow \mathbf{V}_{batch}[k]$
22:     Calculate standardized weights:
23:         $w_i = \frac{J(\theta_i) - \mathbb{E}[\mathcal{J}]}{\sqrt{\text{Var}[\mathcal{J}] + \delta}}$
24:     Estimate optimal descent direction (Gradient Approximation):
25:         $\nabla_\mu J \approx \frac{1}{N} \sum_{i=1}^N w_i \cdot \epsilon_i$
26:     Update search distribution center:
27:         $\mu \leftarrow \mu + \eta \cdot \nabla_\mu J$
28: **end for**
29: **Return** $\mathbf{v}^*$ {Optimal counterfactual state}

---

## D. Mechanistic Analysis Details

### D.1. Linear Probing

To rigorously quantify the spatial information encoded in frozen representations and verify its generalization capability, we implemented a comprehensive linear probing analysis.

**Classifier Design.** We implemented the linear probes as trainable classification layers (Linear($d_{model}$, 4)) operating on the frozen hidden states of the final token. For each layer, we optimized the projection weights to minimize the cross-entropy loss against ground-truth spatial labels.

**Evaluation Protocol.** To distinguish between genuine geometric understanding, we designed a dual-setting protocol: (1) In-Distribution (ID): We utilized the *Controlled Images* dataset, randomly split into a training set (10%) and a test set (90%). This split evaluates the learnability of spatial concepts within the target domain using limited supervision. (2) Probes were trained on the real-world *COCO_QA* dataset and evaluated on the clean-background *Controlled Images* dataset. This setting verifies that the learned spatial features are transferable and robust to domain shifts.

**Training Implementation.** The probes were optimized using AdamW with a learning rate of $1 \times 10^{-4}$ and a batch size of 1. Training was conducted for 15 epochs and the backbone LVLM parameters remained strictly frozen.

## D.2. Logit Lens

To visualize the layer-wise evolution of spatial semantics, we employed the Logit Lens technique, which projects intermediate hidden states directly into the model's vocabulary space.

**Projection Mechanism.** For a given input image and query, we extracted the hidden state $h^{(l)}$ from the final token position at each decoder layer $l$. We then computed the probability distribution over the vocabulary by passing $h^{(l)}$ through the pre-trained language modeling head $W_{head}$: $P^{(l)}(w) = \text{Softmax}\left(W_{head} \cdot h^{(l)}\right)$. This allows us to observe the model's instantaneous prediction confidence at various depths, revealing where the visual spatial signal is strongest and where it potentially degrades.

**Token Tracking Setup.** We focused our analysis on the decoding step responsible for predicting the specific geometric relationship. To quantify the competition between visual perception and language generation, we tracked the probability trajectories of the ground truth token (e.g., "under") versus the dominant hallucination token (e.g., "on"). This comparison enables us to pinpoint the exact moment of semantic collapse, defined as the layer depth where the probability of the hallucinated token surpasses that of the correct token. This crossover point serves as a critical indicator of where strong language priors override the internal visual evidence. Qualitative visualization examples of these trajectories are provided in Appendix F.3.

## D.3. Selection of Intervention Layers

The selection of specific layers for the GRASP intervention is empirically grounded in the mechanistic insights derived from our Linear Probing (Appendix D.1) and Logit Lens (Appendix D.2) analyses. Our investigation reveals a distinct functional stratification across the decoder's depth. In the initial layers (e.g., layers 0–8), the linear probing accuracy remains suboptimal, indicating that the visual features have not yet been fully transformed into linearly separable spatial semantics. Conversely, the deepest layers (e.g., layers 24–32) are prone to semantic collapse, where the Logit Lens visualization confirms that the model's internal representation becomes dominated by language priors, rendering intervention ineffective. The intermediate layers (e.g., layers 12–16 for LLaVA-1.5) emerge as the critical semantic binding phase, characterized by peak probing accuracy before the onset of linguistic dominance. Therefore, we target this specific window for injection to reinforce the geometric signal at its most robust formation stage. We further empirically validate this selection through a detailed ablation study in Appendix F.2.1, which confirms that intervening in these intermediate layers yields the highest performance gains.

## D.4. Robustness to Prompt Variations

In the primary linear probing analysis presented in Appendix D.1, we utilized the standard QA prompt templates (detailed in Table 5) that explicitly query spatial relations (e.g., "Where is..."). While consistent with our main inference setup, this raises a potential confounding factor: does the model explicitly encode visual geometry, or is the high probing accuracy partly driven by the linguistic cues within the interrogative sentence structure?

To rule out the influence of prompt semantics and verify that the geometric signals are intrinsic to the visual representation, we conducted a controlled probing experiment using a **Minimal Context** format. In this setup, we stripped the input of all interrogative structures and spatial keywords. Instead, we introduced special boundary tokens to delimit the subject and object, formatting the input as:

```
USER: <image>\n[START] "Object_A" [MID] "Object_B" [END] ASSISTANT:
```

This formulation forces the model to encode the relationship between the two objects into the latent representation of the [END] token solely through visual attention, without the guidance of a natural language question.

To adapt the model to these control tokens without altering its pre-trained weights, we applied Low-Rank Adaptation (LoRA) exclusively to the token embedding layer, keeping the transformer backbone frozen. We then trained a linear classifier on top of the hidden state at the [END] token position.

As illustrated in Figure 6, our results demonstrate that even in this minimal-context setting, the intermediate layers exhibit robust spatial discriminability. The ID probing accuracy rapidly ascends to a perfect 100% by Layer 14, mirroring the trend observed in the standard QA setting. This confirms that the geometric signal identified in our main experiments is a fundamental, intrinsic property of the vision-language alignment process that exists independently of specific prompt engineering.

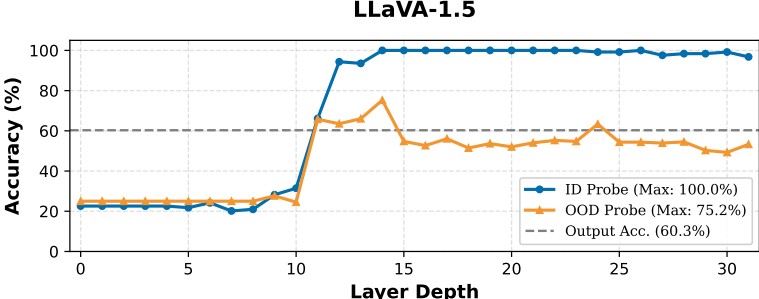

*Figure 6.* **Robustness to Prompt Variations.** We replicate the linear probing analysis using the **Minimal Context** format. The alignment between standard QA and minimal token tagging implies that the spatial signal is not a transient artifact of specific query words, but a fundamental visual primitive that persists robustly regardless of the surface-level linguistic structure.

### D.5. Additional Linear Probing on Other Models

To further verify the generality of our findings, we extended the linear probing analysis to Qwen3-VL, the latest iteration in the Qwen family. As illustrated in Figure 7, the results are consistent with our observations in the main text. This confirms that the competitive dynamics between visual evidence and language biases persist even in the most capable model architectures.

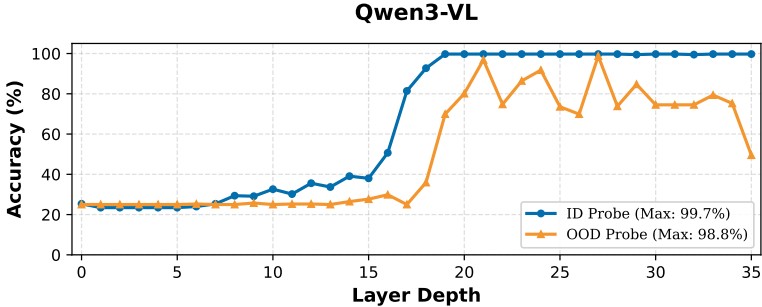

*Figure 7.* **Linear Probing on Qwen3-VL.** Consistent with our main observations, Qwen3-VL exhibits near-perfect spatial discriminability in its intermediate layers. This trend confirms that the representation-output misalignment is a general phenomenon that persists across different model architectures.

## E. Experimental Setup

### E.1. Detailed Prompt Templates

Since standard spatial reasoning benchmarks (WhatsUp, VSR) are originally formulated as image-caption pairs or binary classification tasks, we reformatted them into generative multiple-choice Question-Answering (QA) tasks to evaluate the spatial understanding of LVLMs. This section details the specific prompt templates used for this transformation.

**WhatsUp.** The original WhatsUp dataset consists of image-caption pairs. Following the protocol in AdaptVis, we converted these into a QA format. As shown in Table 5, different subsets utilize different candidate option sets based on the complexity of the scene.

**VSR.** The original VSR benchmark evaluates models on calculating the probability of a caption being True or False. To assess the generative capability of our models, we leveraged GPT-4o to restructure these into a *Yes/No* QA format, requiring the model to explicitly generate the judgment.

**VSI-Bench.** For the video spatial reasoning task, we utilized the standard default prompts provided by the VSI-Bench framework without modification.

*Table 5.* **Prompt templates for benchmark transformation.** Exact formatting used to convert standard benchmarks into generative QA tasks. The `<image>` tag indicates visual embedding placement; for WhatsUp, candidate options vary based on subset complexity.

| Dataset | Subset | Prompt Template | Option Set |
|---|---|---|---|
| **WhatsUp** | Controlled_A | `<image>\nUSER: Where is the <objA> in relation to the <objB>? Answer with left, right, on or under.\nASSISTANT:` | {Left, Right, On, Under} |
| | Controlled_B | `<image>\nUSER: Where is the <objA> in relation to the <objB>? Answer with left, right, front or behind.\nASSISTANT:` | {Left, Right, Front, Behind} |
| | COCO_one | `<image>\nUSER: Where is the <obj> in the photo? Answer with left, right, top or bottom.\nASSISTANT:` | {Left, Right, Top, Bottom} |
| | COCO_two | `<image>\nUSER: Where is the <objA> in relation to the <objB>? Answer with left, right, above, or below.\nASSISTANT:` | {Left, Right, Above, Below} |
| | VG_one | `<image>\nUSER: Where is the <obj> in the photo? Answer with left, right, front, behind, top or bottom.\nASSISTANT:` | {L, R, F, B, Top, Bot} |
| | VG_two | `<image>\nUSER: Where is the <objA> in relation to the <objB>? Answer with left, right, front, behind, above or below.\nASSISTANT:` | {L, R, F, B, Abv, Bel} |
| **VSR** | / | `<image>\nUSER: Determine whether the description about the spatial relationship is correct or not. Answer with yes or no: <Caption>. ASSISTANT:` | {Yes, No} |

## E.2. Detailed Hyperparameters

We provide the comprehensive hyperparameter configurations used for the GRASP framework to facilitate reproduction. The specific intervention layers and strength coefficient ($\alpha$) were determined based on performance on a held-out validation set. For Image Models, we utilized 10% of the *Controlled_A* subset. For Video Models, we utilized 10% of the *Relative Direction* task (Medium difficulty split) from the VSI-Bench.

*Table 6.* **GRASP Inference Hyperparameters.** We report the optimal intervention layers and strength ($\alpha$) for each model architecture. The search parameters ($\sigma, N, T$) are kept consistent across all models.

| Model Architecture | LLaVA-1.5-7B | Qwen2.5-VL-3B | LLaVA-NeXT-Video | Qwen3-VL-7B |
|---|---|---|---|---|
| **Modality** | Image | Image | Video | Video |
| **Positional Encoding** | 1D APE | 2D RoPE | 1D APE | 3D RoPE |
| **Intervention Layers** | [12, 16] | [16, 20] | [9, 16] | [13, 26] |
| **Strength** ($\alpha$) | 1.0 | 0.3 | 0.7 | 0.8 |
| **Search Radius** ($\sigma$) | 15.0 | 15.0 | 15.0 | 15.0 |
| **Sampling Candidates** ($N$) | 2 | 2 | 2 | 2 |
| **Iterations** ($T$) | 2 | 2 | 2 | 2 |

# F. Additional Experimental Results

## F.1. Detailed Performance on VSR

In addition to the WhatsUp benchmark reported in the main text, we provide a detailed evaluation on the Visual Spatial Reasoning (VSR) benchmark. Unlike the generative questions in WhatsUp, VSR presents a distinct challenge by requiring models to verify the truthfulness of complex spatial clauses (e.g., "The cat is touching the table") based on visual evidence. This zero-shot setting strictly tests whether the optimized geometric signal generalizes to diverse linguistic formulations

beyond simple prepositions. As detailed in Table 7, GRASP delivers consistent improvements across both LLaVA and Qwen architectures. Despite the task's heavy linguistic dependency and the binary classification nature, the consistent gains across different metrics confirm that enhancing the intrinsic geometric signal positively transfers to complex verification tasks, sharpening the model's perception of spatial relations.

*Table 7.* **Results on the VSR benchmark.** We report accuracy and F1 score. *Vanilla* denotes the original base models. GRASP consistently improves performance across LLaVA and Qwen architectures, demonstrating generalization to complex spatial semantics.

| Model | Method | Accuracy | F1 Score |
|---|---|---|---|
| **LLaVA-1.5** | Vanilla | 62.4 | 51.3 |
| | **GRASP (Ours)** | **64.7** ↑2.3 | **60.1** ↑8.8 |
| **Qwen2.5-VL** | Vanilla | 70.4 | 72.0 |
| | **GRASP (Ours)** | **72.0** ↑1.6 | **75.5** ↑3.5 |

## F.2. Ablation Study

### F.2.1. SENSITIVITY TO INTERVENTION LAYERS

To validate our hypothesis that spatial semantic alignment occurs primarily in the intermediate layers, we conducted a fine-grained ablation study by varying the depth at which the geometric signal is injected. We divided the decoder layers into blocks of 4 and evaluated the performance on the *Controlled_A* subset.

As illustrated in Figure 8, the two models exhibit distinct sensitivity patterns that correlate directly with their positional encoding paradigms. LLaVA-1.5 (1D APE) displays a pronounced *Inverted-U* sensitivity profile. Because APE injects spatial coordinates only once via the input projection, early visual features lack sufficient semantic grounding; thus, early interventions disrupt raw features. The accuracy surges dramatically to a peak of 86.4% in layers 12–16, the critical window for vision-language semantic binding, before sharply declining in deeper layers as the initial geometric signal decays and language priors dominate. Conversely, Qwen2.5-VL (2D RoPE) explicitly injects spatial priors into the self-attention mechanism at every layer. This recurring, layer-wise operation inherently constructs a highly resilient spatial manifold, rendering its early layers significantly less sensitive to intervention. Consequently, it demonstrates high robustness, maintaining an accuracy above 93% across all depths. However, it still reaches its peak accuracy of 97.6% in layers 16–20, reinforcing our conclusion that despite architectural differences, the intermediate layers universally remain the optimal locus for manipulating spatial representations.

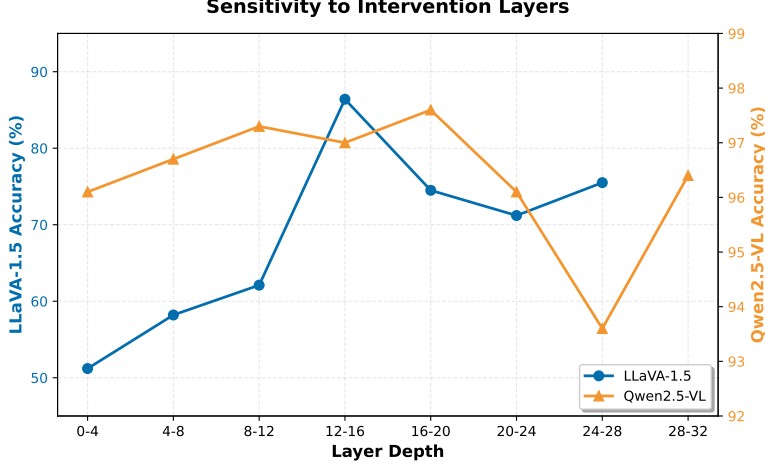

*Figure 8.* **Impact of Intervention Layers.** We compare the performance of LLaVA-1.5 and Qwen2.5-VL when injecting the geometric signal at different decoder depths (grouped by 4 layers). LLaVA-1.5 shows a sharp performance peak in the intermediate layers (12–16), validating our choice of intermediate intervention. Qwen2.5-VL is more robust but consistently favors the 16–20 layer range.

## F.2.2. ABLATION ON INTERVENTION TARGETS

To further validate the rationale behind targeting the self-attention mechanism in our Implicit Trajectory Correction (ITC), we conduct an ablation study comparing interventions on different transformer sub-layers.

As discussed in our methodological formulation, the attention mechanism inherently computes pairwise spatial relationships driven by positional encodings, making it the natural physical receptor for geometric rectification. Fundamentally, self-attention acts as a geometric operator, whereas Feed-Forward Networks (FFNs) primarily function as point-wise semantic dictionaries.

To empirically verify this, we compare the effects of injecting the exact same ITC signal exclusively into the FFN versus the Self-Attention layer on the *Controlled_A* subset using both LLaVA-1.5 and Qwen2.5-VL. As shown in Table 8, forcing topological shifts into the FFN yields sub-optimal gains over the Vanilla baseline, as it disrupts the learned feature distributions and introduces semantic noise. Conversely, targeting the Self-Attention mechanism substantially outperforms the FFN intervention, achieving optimal accuracy gains across both architectures. This empirical result definitively confirms the necessity of isolating our geometric intervention to the self-attention matrices.

*Table 8.* **Ablation on intervention targets.** Empirical comparison between Self-Attention and FFN interventions on the *Controlled_A* subset.

| Intervention Target | LLaVA-1.5 | Qwen2.5-VL |
|---|---|---|
| Vanilla | 60.3 | 90.0 |
| FFN | 69.7 | 90.6 |
| **Self-Attention** | **83.9** | **93.3** |

## F.2.3. SENSITIVITY TO MANIFOLD SEARCH HYPERPARAMETERS

The Manifold Differential Search is governed by three key hyperparameters: the search radius $\delta$, the number of candidate samples $N$, and the optimization steps $T$. To verify the robustness and efficiency of GRASP, we analyze the impact of these parameters on the *Controlled_A* subset using LLaVA-1.5 7B. Unless otherwise specified, the default configuration is set to search radius $\delta = 15°$, candidate samples $N = 2$, and optimization steps $T = 2$.

**Impact of Search Radius ($\delta$).** We varied the maximum rotation limit $\delta$ from $5°$ to $30°$, while fixing $N = 2$ and $T = 2$. To ensure global exploration and avoid local optima, we set the initial search angle to $180°$ before constraining it within the refined radius. As shown in Table 9, performance improves rapidly as $\delta$ increases from $5°$ to $10°$, enabling the model to find sufficient geometric counterfactuals. Crucially, the accuracy forms a broad plateau between $10°$ and $30°$, indicating that GRASP is highly robust to the choice of $\delta$. We adopt $\delta = 15°$ as the default to balance perturbation magnitude and semantic safety.

*Table 9.* **Robustness to Search Radius.** Accuracy stabilizes across a wide range of angles ($5°$–$30°$), with an optimal peak at $\delta = 15°$.

| Radius ($\delta$) | 5° | 10° | **15°** | 20° | 25° | 30° |
|---|---|---|---|---|---|---|
| **Accuracy (%)** | 78.8 | 85.9 | **86.4** | 86.2 | 83.9 | 85.5 |

**Impact of Sampling Candidates ($N$).** Next, we examine the number of candidate samples $N$ used for estimating the optimal direction on the manifold, with $\delta = 15°$ and $T = 2$. Unlike heuristic evolution strategies that often require large populations to explore high-dimensional spaces, our method operates on a structured Lie Group manifold where the local geometry is well-conditioned. We formulate the search as a **Zero-order Gradient Estimation** on the tangent space. As shown in Table 10, while a single sample ($N = 1$) provides a baseline improvement, increasing to $N = 2$ yields a significant gain (+4.8%), confirming that paired sampling provides a critical gradient signal rather than random noise. Crucially, this performance saturation at $N = 2$ incurs negligible memory overhead ($\times 1.04$) when evaluated in FP16 precision, allowing us to maximize efficiency on consumer-grade hardware.

**Efficiency and Convergence Analysis (Steps $T$).** Finally, we evaluate the trade-off between the optimization steps $T$, performance gain, and inference latency (with $\delta = 15°$, $N = 2$). As shown in Table 11, the geometric optimization exhibits rapid convergence: the accuracy jumps significantly from 60.3% to 84.8% in the first step ($T = 1$) and stabilizes at 86.4% by the second step ($T = 2$). While GRASP introduces a latency overhead ($\times 1.7$ at $T = 2$), this cost is justified by the

*Table 10.* **Sampling Efficiency.** Accuracy peaks at minimal candidate samples ($N = 2$), confirming the high quality of the manifold gradient estimation. Memory cost (FP16) is marginal compared to the vanilla model.

| Candidate Samples ($N$) | 0 (Vanilla) | 1 | 2 | 4 |
|---|---|---|---|---|
| Accuracy (%) | 60.3 | 81.6 | **86.4** | 86.4 |
| Peak Memory (GB ↓) | 13.9 | 14.3 (×1.03) | 14.5 (×1.04) | 15.0 (×1.08) |

massive performance improvement (+26.1%). The convergence at $T = 2$ confirms that our method locates the optimal counterfactual efficiently without requiring prolonged iterative search.

*Table 11.* **Efficiency Trade-off.** Performance converges rapidly at $T = 2$. The latency overhead is acceptable given the substantial gain in spatial reasoning accuracy.

| Steps ($T$) | 0 (Vanilla) | 1 | 2 | 3 | 4 |
|---|---|---|---|---|---|
| Accuracy (%) | 60.3 | 84.8 | **86.4** | 86.5 | 86.4 |
| Latency (ms/token ↓) | 37.6 | 56.4 (×1.5) | 64.3 (×1.7) | 77.6 (×2.1) | 82.3 (×2.2) |

**Overall Robustness Summary.** Across all hyperparameter sweeps, GRASP consistently demonstrates superior robustness. Even under suboptimal configurations (e.g., $N = 1$ or $T = 1$), the method achieves an accuracy exceeding 80%, far surpassing the vanilla model (60.3%). This broad operational sweet spot confirms that GRASP relies on the fundamental geometric properties of the representation manifold rather than precise hyperparameter tuning, ensuring stable deployment in diverse real-world scenarios.

### F.3. Visualization of Semantic Collapse

To intuitively understand the mechanism of spatial hallucination, we visualize the layer-wise probability trajectories of spatial tokens using the Logit Lens technique. Figure 9 presents four diverse qualitative examples from the LLaVA-1.5 model, covering different spatial relationships.

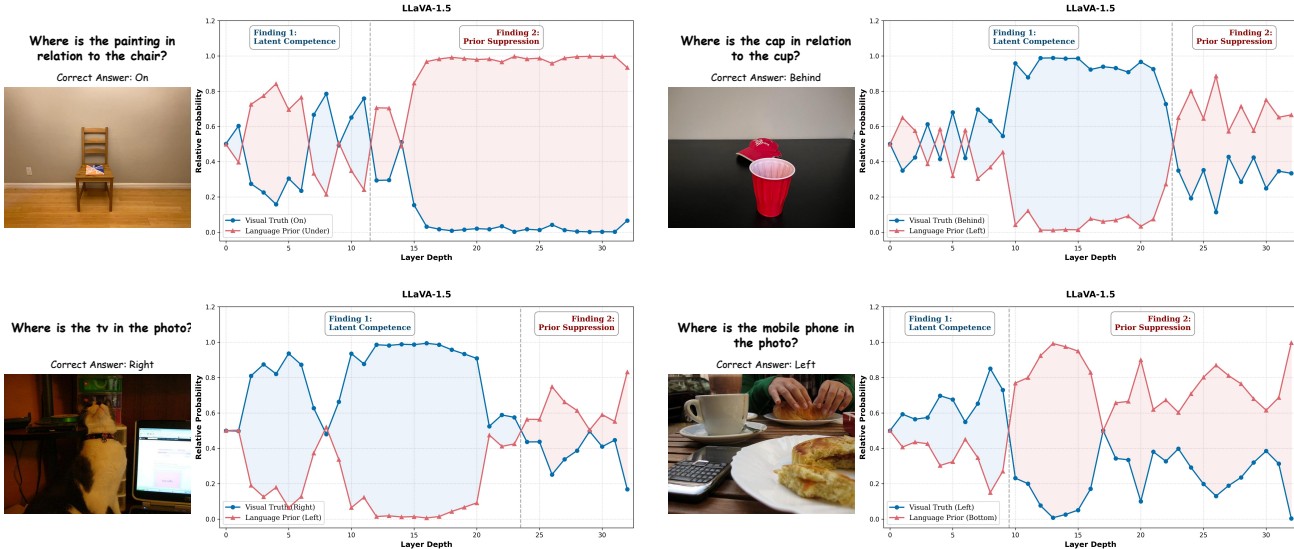

*Figure 9.* **Visualization of Semantic Collapse via Logit Lens.** We track the relative probability evolution of the ground truth token (Blue) versus the hallucinated token (Red) across decoder layers. A consistent *Cognitive Crossover* pattern is observed: the model accurately perceives the visual spatial relationship in intermediate layers (Finding 1), but this correct signal is aggressively suppressed by language priors in deep layers (Finding 2), leading to a final hallucinated output.

**Competitive Dynamics between Visual Evidence and Language Priors.** As illustrated in Figure 9, a consistent competitive pattern emerges across all samples:

- **Phase 1: Latent Competence (Layers 5–18).** In the intermediate layers, the probability of the **Visual Truth** token

(Blue line) rises significantly, often nearing 1.0 (e.g., the "Right" token in the TV example). This confirms that the visual encoder and early projection layers have successfully extracted the correct geometric features. The model *knows* the correct answer internally.

- **Phase 2: The Crossover Point (Layers 18–22).** A critical phase transition occurs where the confidence in the visual truth begins to erode. Simultaneously, the **Language Prior** token (Red line), often a spatial opposite or a statistically frequent preposition, starts to surge. This crossover point marks the onset of semantic collapse.

- **Phase 3: Prior Suppression (Layers 22–32).** In the deep decoder layers, the hallucinated token completely dominates the distribution. For instance, in the "Painting vs. Chair" example (Top-Left), despite the model initially assigning high probability to "On", the final layers override this with "Under". This creates the representation-output misalignment: the internal geometric signal is silenced by the language model's inherent biases.

**Justification for GRASP.** These visualizations provide strong empirical support for our **Implicit Trajectory Correction (ITC)** strategy. Since the geometric signal is strongest and most distinct in the intermediate window (Finding 1), injecting the rectified manifold signal at this specific stage effectively reinforces the "Blue curve" before the "Red curve" takes over. This prevents the crossover event, ensuring that the visual truth survives the deep-layer suppression to reach the final output.

## G. Qualitative Case Studies

To provide an intuitive understanding of GRASP's practical impact, we present representative qualitative examples in Figure 10.

As illustrated, baseline LVLMs (LLaVA-1.5 and Qwen2.5-VL) frequently hallucinate spatial relationships due to strong language priors or contextual biases. By actively rectifying the intermediate geometric manifold, GRASP successfully suppresses this hallucination and accurately identifies the true spatial relation based on the actual visual projection. This demonstrates GRASP's robust capability to awaken latent spatial perceptions.

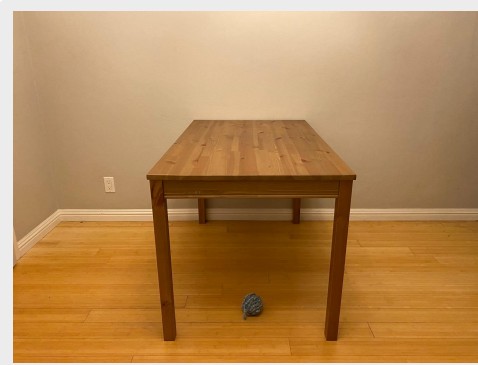

**Prompt:** Where is the ball of yarn in relation to the table? Answer with left, right, on or under.

**LLaVA-1.5:** Left

**Qwen2.5-VL:** The ball of yarn is on the floor to the left of the table.

**Ours:** Under

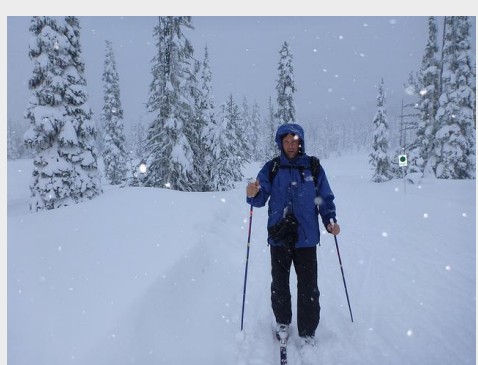

**Prompt:** Where is the person in the photo? Answer with left, right, top or bottom.

**LLaVA-1.5:** Bottom

**Qwen2.5-VL:** Bottom

**Ours:** Right

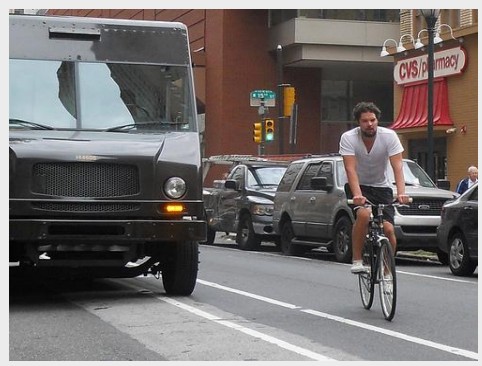

**Prompt:** Where are the cars in relation to the truck? Answer with left, right, front, behind, above or below.

**LLaVA-1.5:** Behind

**Qwen2.5-VL:** The cars is behind of the truck.

**Ours:** Right

*Figure 10.* **Qualitative Examples of GRASP.** Comparison of spatial reasoning outputs. GRASP effectively overrides strong language priors and aligns the output with the true visual geometry.

