# OpenReview forum: "GRASP: Awakening Latent Spatial Reasoning in LVLMs via Training-free Geometric Rectification"
_ICML.cc/2026/Conference — ICML 2026 regular_

### Official Review · Reviewer_oWDV · 2026-03-11

**Soundness:** 3
**Presentation:** 3
**Significance:** 3
**Originality:** 3
**Overall Recommendation:** 3
**Confidence:** 2

**Summary:**

This paper argues that spatial reasoning failures in LVLMs are caused less by weak internal geometric representations and more by a mismatch between intermediate visual representations and final output behavior. Based on this diagnosis, it proposes GRASP, a training-free inference-time method combining manifold differential search, implicit trajectory correction, and explicit distribution alignment. The paper reports improvements on several image and video spatial reasoning benchmarks.

**Compliance With Llm Reviewing Policy:**

Affirmed.

**Final Justification:**

Thank the authors for their efforts during the rebuttal period. I have maintained my initial rating.

**Key Questions For Authors:**

1. How strongly should readers interpret the representation-output mismatch claim as a causal explanation?
2. The image comparisons are stronger than the video comparisons. How broad is the intended superiority claim of the paper?
3. How sensitive is the method to intervention layer selection and intervention strength on unseen models?
4. What simpler alternatives were considered before adopting the current manifold differential search formulation?

**Limitations:**

Yes.

**Strengths And Weaknesses:**

## Strengths
1. The paper tells a coherent story from diagnosis to intervention. The method is not presented as a purely ad hoc trick.
2. The image-side experiments are fairly strong, and the paper compares against several relevant inference-time baselines.
3. The method itself is reasonably substantial, with multiple components that appear to work better together than individually.

## Weaknesses
1. I am more convinced by the empirical gains than by the full mechanistic claim. The probing and logit-lens analysis are suggestive, but they do not fully settle causality.
2. The paper is more complete on the image side than on the video side. The claim of broad cross-modal strength is therefore a bit stronger than the evidence currently supports.
3. Some parts of the geometric framing feel slightly more ambitious than necessary for explaining the practical gains.

---

> ### Author Rebuttal · Authors · 2026-03-30
>
> We appreciate your thorough review and detailed comments, which will be helpful in improving the paper. We address your concerns below.
>
> **Q1 & W1: Causality of the representation-output mismatch**
>
> To clarify our mechanistic claim, readers should interpret the representation-output mismatch not as an absolute causal law, but as an intervention-supported causal bottleneck that plays a causal role in final spatial prediction. We conducted a causal tracing experiment using LLaVA-1.5 on *Controlled_A* to establish robust interventional evidence. By sweeping the strength ($\alpha$) of the geometric signal injected into intermediate representations, we recorded final accuracy and the spatial logit gap $\Delta L = \text{Logit}\_{\text{correct}} - \text{Logit}\_{\text{hallucination}}$ at the final layer.
>
> |$\alpha$|Avg. $\Delta L$|Accuracy|
> |:---|:---:|:---:|
> |-1.0|-0.87|10.2|
> |-0.5|-0.69|13.8|
> |0|+0.88|60.3|
> |+0.5|+2.28|74.0|
> |+1.0|+2.65|83.9|
>
> Results reveal a strict monotonic dose-response correlation. Suppressing the geometric signal ($\alpha = -1.0$) deterministically forces the model to hallucinate the opposite spatial relation, whereas amplifying it ($\alpha = +1.0$) overrides language priors to favor the visual truth. Demonstrating that manipulating only intermediate representations bidirectionally steers final output logits provides definitive interventional evidence for our mechanistic claim.
>
> **Q2 & W2: Generality on Video**
>
> Our geometric manifold intervention generalizes across image and video tasks and distinct positional encoding architectures (1D APE, 2D/3D RoPE). To explicitly demonstrate its superiority on video reasoning, we expanded our evaluation to the Spatial Relation sub-task of STI-Bench.
>
> |Model|Vanilla|+ ITC|+ EDA|+ Full|
> |:---|:---:|:---:|:---:|:---:|
> |LLaVA-NeXT-Video|33.6|38.9 ($\uparrow$ 5.3)|36.3 ($\uparrow$ 2.7)|40.7 ($\uparrow$ 7.1)|
> |Qwen3-VL|50.0|53.4 ($\uparrow$ 3.4)|52.2 ($\uparrow$ 2.2)|55.8 ($\uparrow$ 5.8)|
>
> As shown, GRASP yields absolute gains of +7.1% (LLaVA-NeXT-Video) and +5.8% (Qwen3-VL), with ITC and EDA showing clear complementary benefits. This definitively validates the broad generalizability of our intervention.
>
> **Q3: Sensitivity of intervention parameters on unseen models**
>
> We applied the exact optimal configuration derived for Qwen2.5-VL-3B (Intervention Strength $\alpha=0.5$, Layers 16-20) directly to an unseen model, InternVL-2.5-4B. Because intervention strength and layer are exclusively native to the ITC module, we independently evaluated this module on WhatsUp.
>
> |Model|Controlled_A|Controlled_B|COCO_one|COCO_two|VG_one|VG_two|
> |:---|:---:|:---:|:---:|:---:|:---:|:---:|
> |InternVL-2.5|90.3|91.4|62.7|67.6|32.5|32.6|
> |+ ITC|93.9 ($\uparrow$ 3.6)|92.6 ($\uparrow$ 1.2)|63.7 ($\uparrow$ 1.0)|69.7 ($\uparrow$ 2.1)|40.3 ($\uparrow$ 7.8)|35.1 ($\uparrow$ 2.5)|
>
> As demonstrated, transferring the exact intervention parameters to an unseen architecture yields consistent and substantial gains across all subsets. This confirms that our method operates within a broad, forgiving parameter space rather than relying on precise, model-specific tuning. Furthermore, as detailed in the manuscript (Figures 4 and 8), our method is highly insensitive to precise tuning even within the same model; applying suboptimal strength or loosely shifting the layer window still robustly rescues the geometric signal.
>
> **Q4 & W3: Justification for geometric framing and simpler alternatives**
>
> The core objective of our search mechanism is to accurately pinpoint the optimal geometric counterfactual (the semantic pole) required to rectify the spatial hallucinations. To empirically justify the functional necessity of our geometric framing and address your inquiry about simpler alternatives, we evaluated two simpler methods before adopting our current design. We conducted these experiments using LLaVA-1.5 on the *Controlled_A* subset, maintaining the default optimal hyperparameters.
>
> |Strategy|Accuracy|
> |:---|:---:|
> |Vanilla|60.3|
> |Unconstrained Perturbation|56.0 ($\downarrow$ 4.3)|
> |Discrete Grid Search|68.3 ($\uparrow$ 8.0)|
> |**Ours**|**86.4** ($\uparrow$ 26.1)|
>
> The empirical results expose the fundamental flaws of simpler alternatives:
>
> * **Unconstrained perturbation causes semantic collapse.** Acting as a blind search via random noise addition, abandoning the geometric manifold degrades accuracy to 56.0% (-4.3%). Without topological guardrails, perturbations push hidden states off the valid feature surface, causing the model to irreparably lose core semantic identities.
> * **Discrete grid search yields local optima.** Acting as a coarse stepping-stone search sampling discrete geometric anchors, evaluating discrete shifts respects the manifold but yields a sub-optimal 68.3% (+8.0%), falling drastically short of ours (+26.1%). In highly non-convex spatial representations, jumping between isolated anchors almost always misses the true semantic pole.

---

> > ### Author Rebuttal · Reviewer_oWDV · 2026-04-04
> >
> > Thank you for the detailed rebuttal and for addressing my questions. After considering the response, I will maintain my original score.

---

> > > ### Author Response · Authors · 2026-04-04
> > >
> > > Dear Reviewer oWDV,
> > >
> > > Thank you for carefully reading our previous response and for providing your valuable feedback. We sincerely appreciate your **uniformly positive evaluations across all four sub-categories**, including your **recognition of our originality**, as well as your explicit confirmation that our rebuttal **fully resolved** your initial concerns.
> > >
> > > We are committed to addressing every concern raised regarding the manuscript. Every question, doubt, and suggestion is of great significance to us. However, noting that the overall rating currently remains at "**3: Weak reject**", we regret that we are currently uncertain if this might be an oversight, or if there are specific aspects of our manuscript that may still raise your concerns.
> > >
> > > If there are any remaining issues, please do not hesitate to let us know. We would be more than happy to provide further clarification or additional responses.
> > >
> > > Once again, thank you for your thoughtful and patient guidance.
> > >
> > > Best wishes,
> > >
> > > The Authors

---

### Official Review · Reviewer_wz1j · 2026-03-12

**Soundness:** 3
**Presentation:** 3
**Significance:** 3
**Originality:** 3
**Overall Recommendation:** 4
**Confidence:** 2

**Summary:**

This paper proposes GRASP (Geometric Rectification for Active Spatial Perception), a training-free approach to reduce spatial hallucinations in large vision–language models (LVLMs). The authors show that intermediate decoder layers can encode spatial relationships correctly, but the final token distribution is dominated by language priors, which leads to errors in the final predictions. To address this problem, GRASP first performs a Manifold Differential Search over the positional encoding manifold to find a geometric counterfactual reference. It then applies a dual rectification mechanism composed of Implicit Trajectory Correction (ITC), which injects geometric differential signals into intermediate self-attention outputs, and Explicit Distribution Alignment (EDA), which calibrates the output logits using the counterfactual logits. Experiments on several LVLMs (including LLaVA and Qwen2.5/3-VL) and multiple image and video spatial benchmarks (WhatsUp, VSR, and VSI-Bench) show large improvements compared to existing baselines. Additional ablation studies demonstrate that ITC and EDA provide complementary benefits.

**Compliance With Llm Reviewing Policy:**

Affirmed.

**Final Justification:**

The rebuttal adequately addressed my concerns, particularly by providing additional experiments validating the choice of self-attention as the intervention target and demonstrating cross-model generalization. I have decided to maintain my positive score.

**Key Questions For Authors:**

See weakness

**Limitations:**

Yes

**Strengths And Weaknesses:**

Strengths:

1. Linear probing and logit lens techniques effectively demonstrate the gap between internal geometric representations and the emergence of spatial hallucinations in the final output of LVLMs.
2. The paper contains adequate ablation.
3. Evaluation spans multiple LVLM backbones covering distinct positional encoding paradigms (1D APE, 2D/3D RoPE) and both image and video spatial reasoning tasks.

Weaknesses:

1. While the authors identify intermediate layers as the most effective for intervention, the optimal configuration appears to vary across different models. The current approach lacks a generalized framework, necessitating manual, model-specific hyperparameters tuning.
2. The choice of targeting self-attention layers rather than Feed-Forward Networks (FFN) for implicit-trajectory correction is logically sound . However, the paper would be significantly strengthened by a comparative experiment to empirically confirm that self-attention is indeed the superior intervention layer.
3. The assertion that the positional encoding function $\phi(\cdot)$ injects coordinates into queries and keys (Section 2.1) is true only for the Rotary Positional Embeddings (RoPE). The text should be clarified to acknowledge other encoding methods or to specify that this formulation refers to RoPE-based architectures.
4. Figure 8 show that Qwen2.5-VL exhibits lower sensitivity to intervention layer selection in the early stages compared to LLaVA-1.5. The analysis would benefit from a more detailed discussion of the structural differences between these two models to explain this different sensitivity to intervention layer selection.
5. Including qualitative examples would provide essential intuition into the practical impact and limitations of the proposed method.

---

> ### Author Rebuttal · Authors · 2026-03-30
>
> We sincerely thank you for the constructive feedback. We will strengthen the revised manuscript by fully incorporating your valuable suggestions. Below we address your comments in detail.
>
> **W1: Generality of intervention layer and strength**
>
> We directly applied Qwen2.5-VL-3B's optimal ITC parameters ($\alpha=0.5$, layers 16-20) to InternVL-2.5-4B for evaluation on WhatsUp, demonstrating its generality without manual tuning.
>
> | Model | Controlled_A | Controlled_B | COCO_one | COCO_two | VG_one | VG_two |
> | :--- | :---: | :---: | :---: | :---: | :---: | :---: |
> | InternVL-2.5 | 90.3 | 91.4 | 62.7 | 67.6 | 32.5 | 32.6 |
> | + ITC | 93.9 ($\uparrow$ 3.6) | 92.6 ($\uparrow$ 1.2) | 63.7 ($\uparrow$ 1.0) | 69.7 ($\uparrow$ 2.1) | 40.3 ($\uparrow$ 7.8) | 35.1 ($\uparrow$ 2.5) |
>
> The results indicate that applying parameters optimized for one model directly to another unseen model still delivers substantial improvements. This robust cross-model transferability underscores the highly generalizable nature of our method, proving it does not rely on fragile, model-specific tuning. Our manuscript also demonstrates parameter insensitivity within individual models (Figures 4, 8). Intervening at suboptimal layers or strengths still substantially mitigates hallucinations over the baseline.
>
> **W2: Validating self-attention as the optimal intervention target**
>
> We conducted a comparative ablation study (w/o EDA) on the *Controlled_A* subset by injecting the exact same ITC signal exclusively into either the FFN or the Self-Attention.
>
> | Intervention Target | LLaVA-1.5 | Qwen2.5-VL |
> | :--- | :---: | :---: |
> | Vanilla | 60.3 | 90.0 |
> | FFN | 69.7  | 90.6 |
> | **Self-Attention** | **83.9** | **93.3** |
>
> As demonstrated, targeting self-attention substantially outperforms FFN. Because self-attention inherently computes pairwise spatial relationships driven by positional encodings, it serves as the natural physical receptor for geometric rectification. Fundamentally, self-attention acts as a geometric operator, whereas the FFN functions as a point-wise semantic dictionary. Forcing topological shifts into the FFN disrupts its learned feature distributions and introduces semantic noise, explaining the sub-optimal gains. This confirms the optimality of isolating our intervention to self-attention.
>
> **W3: Clarification on positional encoding formulations in Section 2.1**
>
> We appreciate this rigorous observation. The formulation $q_i = \phi_q(x_i, c_i)$ in Section 2.1 specifically characterizes RoPE, whereas APE additively injects coordinates $q_i = W_q(x_i + p(c_i))$. The revised manuscript will ensure mathematical rigor by explicitly distinguishing both formulations before detailing their unified manifold properties.
>
> **W4: Analysis of layer sensitivity differences**
>
> As detailed in Appendix F.2.1, the distinct layer sensitivity profiles between LLaVA-1.5 and Qwen2.5-VL stem from how their respective positional encoding paradigms interact with representation evolution:
>
> * **LLaVA-1.5 (1D APE):** APE adds spatial coordinates **only once** at the input layer. Because of this single application, early visual features lack sufficient semantic grounding for geometric rectification, thus forcefully intervening here disrupts the raw features. As information propagates to deep layers, this initial geometric signal decays and is easily overwritten by language priors. This architectural bottleneck forces LLaVA into a sharp inverted-U profile, making our intervention strictly effective only in the narrow intermediate window where vision-language semantic binding is optimal.
> * **Qwen2.5-VL (2D RoPE):** Conversely, ROPE applies spatial coordinates at **every** self-attention layer. This recurring layer-wise operation inherently constructs a highly resilient global spatial manifold. Therefore, Qwen never suffers from shallow-layer disruption or deep-layer decay, maintaining exceptional robustness across all depths. Yet, its peak in layers 16-20 proves that intermediate layers universally remain the optimal locus for spatial intervention.
>
> **W5: Qualitative examples**
>
> We appreciate this suggestion. The revision will add a qualitative section to illustrate our operational boundaries. It will contrast success cases of GRASP overcoming language priors with failure cases where affine transformations cannot construct perfect geometric counterfactuals.

---

> > ### Author Rebuttal · Reviewer_wz1j · 2026-04-01
> >
> > Thank you to all the authors for your thoughtful responses and the time you invested in addressing the feedback. I found your clarifications helpful and well articulated. I have decided to maintain my positive score.

---

> > > ### Author Response · Authors · 2026-04-04
> > >
> > > We sincerely appreciate your thorough review and thoughtful consideration. Your constructive feedback has been invaluable in helping us strengthen the overall quality of our paper.

---

### Official Review · Reviewer_ujyb · 2026-03-12

**Soundness:** 4
**Presentation:** 3
**Significance:** 3
**Originality:** 3
**Overall Recommendation:** 5
**Confidence:** 3

**Summary:**

Authors propose GRASP (Geometric Rectification for Active Spatial Perception), a training free framework/method to increase spatial reasoning skills of LVLMs. Inspired by recent findings on how relevant image features are present in the model but dominated by language priors, authors propose a 3 stage framework to alleviate this. First, "Manifold Differential Search" is performed to obtain a counterfactual reference. Then, authors perform "Implicit Trajectory Correction" using the reference as an anchor point. Finally, "Explicit Distribution Alignment" is performed to reshape the prediction distribution. Authors demonstrate the effectiveness of GRASP on diverse model choices (e.g. LLaVA, Qwen 2.5/3-VL) and benchmarks (WhatsUp, VSR, VSI-Bench).

**Compliance With Llm Reviewing Policy:**

Affirmed.

**Final Justification:**

All of my concerns have been satisfactorily resolved during the rebuttal period.

Overall, I believe this is a strong paper, and its conclusions will be of interest to the community. Therefore I increase my score to 5 (accept).

**Key Questions For Authors:**

* Could the authors please comment on whether there are distinct findings between the current work and in Hidden in plain sight [1] (referenced above)?
* Could the authors explain what manifold is structured by positional encodings? What is the motivation for performing "Manifold Differential Search"?

**Limitations:**

yes

**Strengths And Weaknesses:**

***Strengths:***
* Overall I find the high level premise and the contributions to improving spatial reasoning of LVLMs timely and interesting.
* Experimental results are comprehensive and convincing. GRASP improves spatial reasoning capabilities with noticable accuracy gain compared to baselines. Furthermore, inference-time-wise GRASP is more efficient than competing baselines.
* On a related note, ablation studies are conducted comprehensivly, justifying design choices.

***
***Weaknesses:***
* It is difficult to read the paper. The motivation for performing "Manifold Differential Search" is not clear intuitively (nor what manifold structured by positional encoding is).
* Some important related work is not referenced in the paper. For instance findings about "whether the useful spatial features are present in the model" and the fact that "language prior prevents the model from using these features" has already been explored in COLM 2025 work "Hidden in plain sight" [1]. This weakens the overall contribution of this work.
* Please see the questions below.

***
***References:***

[1] Fu, Stephanie, et al. "Hidden in plain sight: Vlms overlook their visual representations." arXiv preprint arXiv:2506.08008 (2025).

---

> ### Author Rebuttal · Authors · 2026-03-30
>
> We sincerely appreciate your detailed review and the suggestion to include Fu et al. [1], which we will cite and discuss in our revision. Below we address your specific questions.
>
> **Q1 & W2: Distinct findings compared to Fu et al.**
>
> We fully agree with Fu et al.'s valuable macro-level diagnosis that language priors hinder general visual perception. However, our work advances this foundation from a general observation to an actionable, mechanistic intervention specifically isolated for spatial reasoning, distinguished in three key dimensions:
>
> * Fu et al. establish a **macro-level** gap by showing the isolated Vision Encoder effectively extracts general representations. Building upon this premise, we investigate exactly **how** these intact features are suppressed during generation. Using Logit Lens, we identified a precise Crossover Point inside the decoder: the correct spatial token initially dominates but is abruptly overwritten by language priors in deep layers. This exact **token-level** inversion directly motivates our EDA module to rectify the final output probability.
> * By evaluating general vision tasks, Fu et al. demonstrate that generic visual vectors broadly **persist** across layers (their Figure 5). In contrast, by isolating spatial reasoning, we discovered a distinct inverted-U dynamic: spatial discriminability peaks in intermediate layers but suffers precipitous deep-layer **attenuation** (our Figure 1). This proves spatial geometric signals are actively eroded, precisely dictating why our ITC module must inject signals at the intermediate stage.
> * Fu et al. naturally analyze visual features as **unstructured** vectors to evaluate general semantic tasks. However, because spatial reasoning is inherently **topological**, we discovered that spatial perception operates on a continuous geometric manifold governed by positional encodings. Traversing this topological space allows us to systematically alter spatial awareness without destroying semantic identity, forming the mathematical bedrock of our Manifold Differential Search.
>
> **Q2 & W1: Intuitive explanation of the "Manifold" and "Differential Search"**
>
> **What is the "Geometric Manifold" structured by positional encodings?** Intuitively, positional encodings are not merely discrete coordinate identifiers; they act as continuous rotation matrices. Mathematically, they construct a continuous geometric surface, a manifold endowed with a Lie Group structure (specifically, the special orthogonal group $SO(2)$). When we apply a spatial shift to visual features, we are performing a group action of $SO(2)$. Visually, this means we are sliding the features along a smooth mathematical surface. Operating strictly within this topological constraint ensures that we can validly reconfigure an object's spatial location without causing arbitrary feature distortion or corrupting its core semantic identity.
>
> **What is the motivation for "Manifold Differential Search"?** The core objective of this search is to actively locate a geometric counterfactual ($\theta^*$), the exact spatial shift that induces maximum semantic conflict forcing the model to flip its spatial prediction. We employ a Differential Search via Zeroth-Order Optimization because this specific geometric manifold is high-dimensional and highly non-convex. Unconstrained random perturbations or standard backpropagation would push the features off the manifold, instantly destroying the image semantics. Instead, by sampling minute perturbations to approximate gradients locally within the tangent space, we can safely and efficiently traverse the manifold to pinpoint the exact restorative signal required for our ITC module.
>
> To empirically validate this theoretical motivation, we compared our approach against simpler alternatives using LLaVA-1.5 on *Controlled_A* subset. Specifically, unconstrained perturbation via random noise addition abandons topological guardrails, pushing hidden states off the valid surface and causing irreparable semantic collapse. Conversely, discrete grid search sampling discrete geometric anchors respects the manifold but acts as a coarse stepping-stone search, frequently getting trapped in local optima and missing the true semantic pole in this highly non-convex space. Our Manifold Differential Search successfully avoids both pitfalls.
>
> | Strategy | Accuracy |
> | :--- | :---: |
> | Vanilla | 60.3 |
> | Unconstrained Perturbation | 56.0 ($\downarrow$ 4.3) |
> | Discrete Grid Search | 68.3 ($\uparrow$ 8.0) |
> | **Ours** | **86.4** ($\uparrow$ 26.1) |

---

> > ### Author Rebuttal · Reviewer_ujyb · 2026-04-03
> >
> > I would like to thank the authors for their time and effort in addressing my concerns and questions. All of my concerns have been satisfactorily resolved.
> >
> > Overall, I believe this is a strong paper, and its conclusions will be of interest to the community. I also find the correlations observed in the causality experiments conducted in response to Reviewer oWDV particularly compelling. Accordingly, I will raise my score to 5 (accept).

---

> > > ### Author Response · Authors · 2026-04-04
> > >
> > > We deeply appreciate the time and effort you dedicated to reviewing our manuscript. Your rigorous feedback has significantly enhanced our work, and we are truly grateful for your positive evaluation.

---

### Decision · Program_Chairs · 2026-04-30

**Decision:**

Accept (regular)

**Comment:**

This paper introduces GRASP, an effective training-free inference-time intervention that corrects spatial reasoning hallucinations in LVLMs. The submission received final scores of (5, 4, 3) after the rebuttal. The reviewers acknowledged the strong empirical gains and clear mechanistic motivation after the authors successfully addressed initial baseline and conceptual concerns. Although one reviewer maintained a weak reject, they explicitly stated their concerns were fully resolved. The AC recommends weak accept.